Journal of Data-centric Machine Learning Research (2024)        Submitted 10/23; Revised 10/24; Published 10/24

# OpenOOD v1.5: Enhanced Benchmark for Out-of-Distribution Detection

**Jingyang Zhang**                                                           *Duke University*
**Jingkang Yang**                                        *S-Lab, Nanyang Technological University*
**Pengyun Wang**                                          *The Australian National University*
**Haoqi Wang**                                                                        *EPFL*
**Yueqian Lin**                                                              *Duke University*
**Haoran Zhang**                                                             *Duke University*
**Yiyou Sun**                                               *University of Wisconsin-Madison*
**Xuefeng Du**                                              *University of Wisconsin-Madison*
**Yixuan Li**                                               *University of Wisconsin-Madison*
**Ziwei Liu**                                            *S-Lab, Nanyang Technological University*
**Yiran Chen**                                                               *Duke University*
**Hai Li**                                                                   *Duke University*

*Code: https://github.com/Jingkang50/OpenOOD/*
*Leaderboard: https://zjysteven.github.io/OpenOOD/*

**Reviewed on OpenReview:** *https://openreview.net/forum?id=cnnTnJQigs*

**Editor:** Yang Liu

## Abstract

Out-of-Distribution (OOD) detection is critical for the reliable operation of open-world intelligent systems. Despite the emergence of an increasing number of OOD detection methods, the evaluation inconsistencies present challenges for tracking the progress in this field. OpenOOD v1 initiated the unification of the OOD detection evaluation but faced limitations in scalability and scope. In response, this paper presents OpenOOD v1.5, a significant improvement from its predecessor that ensures accurate and standardized evaluation of OOD detection methodologies at large scale. Notably, OpenOOD v1.5 extends its evaluation capabilities to large-scale data sets (ImageNet) and foundation models (*e.g.*, CLIP and DINOv2), and expands its scope to investigate full-spectrum OOD detection which considers *semantic* and *covariate* distribution shifts at the same time. This work also contributes in-depth analysis and insights derived from comprehensive experimental results, thereby enriching the knowledge pool of OOD detection methodologies. With these enhancements, OpenOOD v1.5 aims to drive advancements and offer a more robust and comprehensive evaluation benchmark for OOD detection research.

**Keywords:** out-of-distribution detection, open-set recognition, distribution shifts

## 1 Introduction

For intelligent recognition systems to reliably operate in the open world, it is crucial for them to have the capability of detecting and handling unknown inputs. This problem is commonly formulated as Out-of-Distribution (OOD) detection (Hendrycks and Gimpel,

2017) or Open-Set Recognition (OSR, Bendale and Boult, 2016). In the context of image classification, OOD detection seeks to enable the identification of images that do not belong to any of the known, in-distribution (ID) categories of the classifier.

Recent years have witnessed a surge of over hundreds of papers on OOD detection (Yang et al., 2021b). Despite the increasing attention and the importance of this research problem, tracking the progress in this field has been hindered by three *evaluation pitfalls* that are often overlooked by researchers: **1)** confusing terminologies, **2)** inconsistent data sets, and **3)** erroneous practices (we refer readers to Appendix A for a detailed discussion). As a result, the OOD detection community has a pressing need for a unified test platform and benchmark to accurately evaluate current and future methodologies. One work that comes close to this goal is the first version of OpenOOD (Yang et al., 2022a), which yet is limited in *scale* and *scope*. For example, OpenOOD v1's evaluation was mostly performed on small-scale data sets like MNIST (Deng, 2012) and CIFAR (Krizhevsky et al., 2009a,b), while larger data sets such as ImageNet (Deng et al., 2009) obviously carry greater importance (Hendrycks et al., 2022; Wang et al., 2022; Djurisic et al., 2023; Zhang et al., 2023b).

Building upon OpenOOD v1, in this work we present OpenOOD v1.5 which features fair and accurate evaluation of OOD detection on a larger scale and with a broader scope. Concretely, we make the following extensions and contributions.[1]

**Large-scale experiments and results.** In addition to the small data sets included in v1, OpenOOD v1.5 provide the *most* extensive experiment results for nearly 40 methods (and their combinations) on ImageNet-1K, which serve as a comprehensive reference for later works. To facilitate future research in large-scale settings with affordable computational cost, we also introduce a new benchmark constructed with ImageNet-200, a subset of ImageNet-1K. Furthermore, we evaluate two large-scale foundation models (CLIP, Radford et al., 2021 and DINOv2, Oquab et al., 2023) to provide initial inspection of their performance on OOD detection tasks.

**Investigation on full-spectrum detection.** Besides the *standard* setting considered in v1, OpenOOD v1.5 (for the first time) closely evaluates *full-spectrum* OOD detection (Yang et al., 2022b), an important setting that considers OOD *generalization* (Hendrycks and Dieterich, 2019; Hendrycks et al., 2021a) and OOD *detection* simultaneously. Compared with the standard setting which is studied by most existing works, we show that full-spectrum detection poses significant challenge for all current approaches.

**New insights.** With comprehensive results from OpenOOD v1.5, we are able to provide several valuable observations. For example, we identify that there is *no single winner* that always outperforms others across multiple data sets. Meanwhile, we observe that *data augmentations* (Geirhos et al., 2019; Cubuk et al., 2020; Hendrycks et al., 2020, 2021a,c; Pinto et al., 2022) *help* with OOD detection in both standard and full-spectrum setting. Our insights help assess the current state of OOD detection and provide future directions for the community.

---

1. As a benchmarking tool that continuously evolves, OpenOOD v1.5 also introduces several new features and updates that make itself more accessible and easier to use. We describe them in Appendix B.

## 2 Problem Statement

Our work focuses on OOD detection in the context of multi-class image classification, since it is one of the most fundamental and commonly studied problems. Yet, to make our discussion generalizable, we start with a formal mathematical definition of the objective of OOD detection task, which can be applied to each specific (sub-)task, *e.g.*, classification and regression. After that, we introduce the specific problem statement and evaluation metrics by diving into the considered image classification scenario.

**Mathematical definition.** The first thing to define is that given a reference distribution $\mathcal{D}_{\text{ID}}$ (which is considered in-distribution, ID), what data is defined as out-of-distribution. Interestingly, we find that such definition is missing from existing works, including those seminal ones and theoretical ones. Here we propose the following definition:

**Definition 1** *Assume a reference distribution (that is considered in-distribution) $\mathcal{D}_{ID}$, where $(x, y) \sim \mathcal{D}_{ID}$ with $x$ being the input and $y$ being the output/label. Denote the probability density function as $p$. Then a distribution $\mathcal{D}_{OOD}$ is out-of-distribution w.r.t. $\mathcal{D}_{ID}$ if for all $\delta \in (0, 1)$, there exists a region $\mathcal{D}$ such that 1) $\iint_{\mathcal{D}} p_{\mathcal{D}_{OOD}}(x, y) dx dy > \delta$ and 2) $\forall \epsilon > 0$, $\iint_{\mathcal{D}} p_{\mathcal{D}_{ID}}(x, y) dx dy < \epsilon$.*

Essentially, the definition indicates that within each region $\mathcal{D}$ where the probability mass of $\mathcal{D}_{\text{OOD}}$ is significant, the probability mass of $\mathcal{D}_{\text{ID}}$ within that same region is arbitrarily small. A concrete example is that a car image is OOD w.r.t. an animal classification dataset, as under the joint distribution of the animal and its category, the probability density of a car and its label appearing together is zero. Definition 1 generalizes to other data modalities and tasks as well. For instance, for a regression task that predicts the scale of positivity of a movie review, a sentence that talks about something completely irrelevant to movies would be OOD according to the definition. We note that our definition aligns with the high-level yet informal description of OOD data in the survey work of Yang et al. (2021b), where they describe it as "test samples to which the model cannot generalize." In our definition, samples that have arbitrarily small probability density under the reference distribution would be ill-posed for a model that is trained on the reference distribution and thus is certainly something the model cannot make a prediction for.

Now that we have defined the OOD data, we give the formal objective of OOD detection, which is adapted from one of the seminal works (Scheirer et al., 2012). It is originally defined for classification only, and here we generalize it with the help of our Definition 1.

**Definition 2** *Consider a reference distribution $\mathcal{D}_{ID}$ and a distribution $\mathcal{D}_{OOD}$ that is OOD w.r.t. it according to Definition 1. Also consider a detection function $f$ where $f(x) = 0$ when it predicts the input being from $\mathcal{D}_{OOD}$ and $f(x) = 1$ when it predicts the input being from $\mathcal{D}_{ID}$. The objective of OOD detection is defined by finding a detection function $f$ that minimizes the open space risk:*

$$\arg\min_{f} R_{\mathcal{O}}(f), \ \ where \ R_{\mathcal{O}}(f) = \frac{\iint f(x) p_{\mathcal{D}_{OOD}}(x, y) dx dy}{\iint f(x) p_{\mathcal{D}_{OOD}}(x, y) dx dy + \iint f(x) p_{\mathcal{D}_{ID}}(x, y) dx dy}.$$

From Definition 2, we see that the more we label the OOD space as ID ($f(x) = 1$), the greater the open space risk $R_{\mathcal{O}}(f)$. In contrast, an ideal function would give zero open space risk, thus the overall goal of OOD detection is to minimize the open space risk. While Definition 2 gives a concrete mathematical definition, there is some nuance in practice. First, OOD detection often roots upon a base task, whose objective should be considered as well. Second, as OOD detection is a binary classification task, established metrics such as AUROC are more often used rather than the open risk itself.

Upon the discussed mathematical definition, we now proceed to describing the problem statement specifically within the image classification scenario. However, our framework has the potential to generalize to other tasks such as regression, which we demonstrate in Appendix D. We first introduce the two settings of standard and full-spectrum detection, before finally touching the evaluation metrics for the problem.

**Standard OOD detection.** Given an image classification problem, there will always exist a pre-defined set of semantic categories/labels $\mathcal{Y}_{\mathrm{ID}}$, which is considered in-distribution (ID). In an open world, an OOD label space $\mathcal{Y}_{\mathrm{OOD}} = \{y | y \notin \mathcal{Y}_{\mathrm{ID}}\}$ also exists. In the case of a CIFAR-10 classifier, for example, $\mathcal{Y}_{\mathrm{ID}} = \{\texttt{airplane}, \texttt{bird}, ..., \texttt{truck}\}$, and $\mathcal{Y}_{\mathrm{OOD}} = \{\texttt{apple}, \texttt{mountain}, ...\}$. At inference time, for any image $x$ with ground-truth label $y$, an ideal classifier with OOD detection capability should behave in a way that: 1) it can identify whether $y \in \mathcal{Y}_{\mathrm{ID}}$ or $y \in \mathcal{Y}_{\mathrm{OOD}}$, and 2) if $y \in \mathcal{Y}_{\mathrm{ID}}$, it classifies $x$ to one of the ID categories accurately. While the second goal is fundamental for any image classifier, the first goal is the unique focus of OOD detection.

To enable an image classifier $f$ to detect OOD samples, an OOD detector $G$ is needed on top of $f$, which can usually be formulated as

$$G(x; f) = \begin{cases} 1 \ (\text{OOD}), & S(x; f) \geq \lambda \\ 0 \ (\text{ID}), & S(x; f) < \lambda \end{cases}, \tag{1}$$

where $S(\cdot)$ is a scoring function that outputs an score to indicate the "OOD-ness" of each sample, and $\lambda$ is a case-dependent threshold.

**Full-spectrum OOD detection.** Standard OOD detection essentially studies the *semantic* distribution shifts (between $\mathcal{Y}_{\mathrm{ID}}$ and $\mathcal{Y}_{\mathrm{OOD}}$); it yet ignores another type of distribution shifts that are prevalent in real-world, *i.e.*, *covariate* shifts (Hendrycks and Dietterich, 2019; Recht et al., 2019; Hendrycks et al., 2021a). In fact, there are active research endeavors that study the robustness to covariate shifts (often termed as OOD robustness or OOD generalization), especially in applications such as autonomous driving (Kong et al., 2023, 2024a,b). Full-spectrum detection (Yang et al., 2022b) for the first time considers both semantic-shifted ID images (what we call "OOD" images in this work) and covariate-shifted ID images (csID). Note that in the context of OOD detection, csID images are still *in-distribution* since their semantic labels are still within $\mathcal{Y}_{\mathrm{ID}}$ (*e.g.*, a blurry dog image should be recognized as $\texttt{dog}$ nonetheless). A concrete illustration of full-spectrum detection is shown in Figure 1. We will further discuss the ideal behavior of full-spectrum detection in Section 6.2.

**Evaluation metrics.** Recall from Equation 1 that OOD detection is a binary classification problem. Following the convention in machine learning (Provost et al., 1998) and the seminal work of OOD detection (Hendrycks and Gimpel, 2017), we treat the "anomalous" OOD samples as *positive* and the "normal" ID samples as *negative*. We use three established

Figure 1: Illustration of full-spectrum OOD detection (Yang et al., 2022b) using our ImageNet benchmark. Standard detection only concerns *semantic* shift by detecting (c) + (d) from (a), while full-spectrum detection takes into account *covariate* shift and aims to separate (c) + (d) from (a) + (b). An ideal system should be robust to the non-semantic covariate shift (OOD *generalization*) while being able to identify semantic shift (OOD *detection*).

metrics: 1) area under the receiver operating characteristic (AUROC), 2) area under the Precision-Recall curve (AUPR), and 3) false positive rate at 95% true positive rate (FPR@95). AUROC and AUPR are threshold-independent measurements, while FPR@95 reflects the performance at a specific threshold.

## 3 Evaluation Protocol

Based upon the problem statement, we next describe the (abstract) evaluation protocol specified by OpenOOD v1.5, which is designed to ensure the most rigorous evaluation of current and future methodologies.

**Standard OOD detection.** We consider the data set that characterizes the given image classification task as in-distribution (ID) data set $\mathcal{D}_{\mathrm{ID}}$. Following common practices, we take publicly available data sets whose categories are OOD w.r.t. $\mathcal{Y}_{\mathrm{ID}}$ as the source of OOD samples $\mathcal{D}_{\mathrm{OOD}}$. In general, the image classifier will be trained with ID training images[2] $\mathcal{D}_{\mathrm{ID}}^{\mathtt{train}}$ and evaluated with ID test images $\mathcal{D}_{\mathrm{ID}}^{\mathtt{test}}$ and OOD test images $\mathcal{D}_{\mathrm{OOD}}^{\mathtt{test}}$. For each $\mathcal{D}_{\mathrm{ID}}$, we evaluate the classifier and detector with multiple $\mathcal{D}_{\mathrm{OOD}}$ for comprehensiveness. Moreover, we divide the considered OOD data sets into two groups: near-OOD and far-OOD (or equivalently, hard-OOD and easy-OOD) (Ahmed and Courville, 2020). The grouping is based on either image semantics or empirical difficulty, which can give a more fine-grained evaluation of OOD detectors in the face of different OOD samples (see Section 4 for more details).

---

2. One exception is when we evaluate foundation models like CLIP (Radford et al., 2021) and DINOv2 (Oquab et al., 2023), which are pre-trained with large amount of data.

| ID samples | OOD test samples $\mathcal{D}_{\text{OOD}}^{\text{test}}$ | | OOD training samples | OOD validation samples |
|---|---|---|---|---|
| $\mathcal{D}_{\text{ID}}$ | Near-OOD | Far-OOD | $\mathcal{D}_{\text{OOD}}^{\text{train}}$ | $\mathcal{D}_{\text{OOD}}^{\text{val}}$ |
| CIFAR-10 | CIFAR-100, TIN | MNIST, SVHN, Textures, Places365 | TIN-597 | 20-class hold-out set of TIN |
| CIFAR-100 | CIFAR-10, TIN | MNIST, SVHN, Textures, Places365 | TIN-597 | 20-class hold-out set of TIN |
| ImageNet-200 | SSB-hard, NINCO | iNaturalist, Textures, OpenImage-O | ImageNet-800 | Hold-out set of OpenImage-O |
| ImageNet-1K | SSB-hard, NINCO | iNaturalist, Textures, OpenImage-O | N/A | Hold-out set of OpenImage-O |

Table 1: Summary of the 4 standard OOD detection benchmarks of OpenOOD v1.5. All data sets used in our work are either existing public data sets or subsets that we curate from existing ones. Please see Section 4 for details. Full-spectrum benchmarks only adds additional covariate-shifted samples into the ID test set, which we also describe in text.

**Full-spectrum OOD detection.** As mentioned earlier, full-spectrum detection additionally considers covariate-shifted ID samples (csID). In practice, this is done by incorporating csID samples $\mathcal{D}_{\text{csID}}^{\text{test}}$, together with $\mathcal{D}_{\text{ID}}^{\text{test}}$, to serve as the ID test data.

**Validation data for hyperparameter tuning.** Many OOD detectors have tunable hyperparameters. In contrast to earlier works which determine hyperparameter values using test samples $\mathcal{D}_{\text{ID}}^{\text{test}}$ and $\mathcal{D}_{\text{OOD}}^{\text{test}}$ (Liang et al., 2018; Lee et al., 2018; Hsu et al., 2020; Kong and Ramanan, 2021), we instead introduce ID and OOD validation samples $\mathcal{D}_{\text{ID}}^{\text{val}}$ and $\mathcal{D}_{\text{OOD}}^{\text{val}}$ to ensure realistic evaluation and avoid reporting overoptimistic results. Specifically, $\mathcal{D}_{\text{ID}}^{\text{val}}$ is a small subset held out from $\mathcal{D}_{\text{ID}}^{\text{test}}$, and $\mathcal{D}_{\text{OOD}}^{\text{val}}$ is carefully constructed such that $\mathcal{Y}_{\text{OOD}}^{\text{val}} \cap \mathcal{Y}_{\text{OOD}}^{\text{test}} = \varnothing$. We also use these validation samples for selecting the "best" model checkpoint during training.

**OOD training data.** A line of works choose to incorporate OOD images at training time to improve OOD detection capability (Hendrycks et al., 2019a; Yu and Aizawa, 2019; Yang et al., 2021a; Zhang et al., 2023a). To avoid trivial evaluation in such cases (Hendrycks et al., 2019a), we make distinction between OOD training samples $\mathcal{D}_{\text{OOD}}^{\text{train}}$ and OOD test samples $\mathcal{D}_{\text{OOD}}^{\text{test}}$, where they should have non-overlapping categories (*i.e.*, $\mathcal{Y}_{\text{OOD}}^{\text{train}} \cap \mathcal{Y}_{\text{OOD}}^{\text{test}} = \varnothing$).

**Remark.** One important goal of OpenOOD is to provide rigorous and accurate evaluation. This can be reflected by our efforts in preparing validation data and meaningful OOD training data. We notice that such data curation has long been overlooked by many prior works; in fact, even the previous version of OpenOOD (Yang et al., 2022a) lacks some of the data considerations in this work (*e.g.*, $\mathcal{Y}_{\text{OOD}}^{\text{train}}$ overlapped with $\mathcal{Y}_{\text{OOD}}^{\text{test}}$, which silently turned OOD detection into a trivial supervised classification problem in an unrealistic way).

## 4 Supported Benchmarks and Methods

In this section, we introduce the supported benchmarks and methods of OpenOOD v1.5. While we discuss each benchmark in detail in the main body, we leave the expanded description of each implemented methods in Appendix C for conciseness (understanding the technical detail of specific methods is not necessary for interpreting the benchmarking results that we will present later). We start by the 4 benchmarks for *standard* OOD detection; a summary is provided in Table 1. Then we dive into the 2 *full-spectrum* benchmarks that focus on large-scale ImageNet settings. Lastly we go over the supported methods on a high level.

**CIFAR-10.** The first benchmark considers CIFAR-10 (Krizhevsky et al., 2009a) as ID. We use the official train set with 50,000 samples as $\mathcal{D}_{\text{ID}}^{\texttt{train}}$ and hold out 1,000 samples from the test set to form $\mathcal{D}_{\text{ID}}^{\texttt{val}}$, while the remaining 9,000 test samples are taken as $\mathcal{D}_{\text{ID}}^{\texttt{test}}$. The *near*-OOD group contains CIFAR-100 (Krizhevsky et al., 2009b) and Tiny ImageNet (TIN, Le and Yang, 2015). 1,203 images are removed from TIN due to their overlap with CIFAR (Yang et al., 2021a). Another 1,000 TIN images covering 20 categories are held out to serve as $\mathcal{D}_{\text{OOD}}^{\texttt{val}}$ which is disjoint with $\mathcal{D}_{\text{OOD}}^{\texttt{test}}$. The *far*-OOD group consists of MNIST (Deng, 2012), SVHN (Netzer et al., 2011), Textures (Cimpoi et al., 2014), and Places365 (Zhou et al., 2017) with 1,305 images removed due to semantic overlap (Yang et al., 2021a). The OOD group is determined by image content and semantics: Near-OOD images are similar to CIFAR-10 as they all include specific objects (*e.g.*, `bottles`, `apples`, etc.), while far-OOD images are either numerical digits, textural patterns, or scene imagery, which deviate much from ID images in both semantic meaning and low-level statistics (Ahmed and Courville, 2020), thus making themselves easier to be detected.

**CIFAR-100.** The CIFAR-100 benchmark is similar to the CIFAR-10 one. We take 1,000 samples out of the ID test set as ID validation data. The *near*-OOD split is made of CIFAR-10 and TIN. The *far*-OOD group and validation OOD data are the same as in CIFAR-10 case.

**ImageNet-1K**. We use 45,000 images from the ImageNet validation set (Deng et al., 2009) as $\mathcal{D}_{\text{ID}}^{\texttt{test}}$, while the remaining 5,000 images serve as $\mathcal{D}_{\text{ID}}^{\texttt{val}}$. We do not modify the original 1.2M ImageNet training set so that any pre-trained models can be directly evaluated with OpenOOD.

We include SSB-hard (Vaze et al., 2022) and NINCO (Bitterwolf et al., 2023) in the *near*-OOD group for ImageNet-1K. SSB-hard consists of 49,000 images and covers 980 categories selected from ImageNet-21K (Ridnik et al., 2021). NINCO is a new data set of 5,879 images manually curated by Bitterwolf et al. (2023). The *far*-OOD group considers iNaturalist (Van Horn et al., 2018), Textures (Cimpoi et al., 2014), and OpenImage-O (Wang et al., 2022). The first two data sets were first used as benchmarks in the MOS paper (Huang and Li, 2021) and later have become popular for evaluating ImageNet models. OpenImage-O is curated from Open Images (Kuznetsova et al., 2020). 1,763 images from OpenImage-O are picked out as $\mathcal{D}_{\text{OOD}}^{\texttt{val}}$. Unlike CIFAR, ImageNet has 1,000 diverse visual categories, making it hard or ambiguous to define near-OOD and far-OOD based on label semantics. Instead, here we make the categorization by inspecting the empirical performance of OOD detectors on each OOD data set, which reflect how difficult the data set is for the OOD detection task. As will be seen in our results, the margin between the near-OOD and far-OOD detection AUROC is often large, meaning that the two groups indeed present distinct level of difficulty for all detectors.

**ImageNet-200.** We further consider a 200-class subset of ImageNet-1K which is still relatively large yet requires less compute to experiment with. ImageNet-200 has the same 200 categories as ImageNet-R (Hendrycks et al., 2021a). It shares the same OOD data sets as our ImageNet-1K benchmark. The ImageNet-200 benchmark can facilitate the evaluation of scalability as one can make straight comparison between the results on ImageNet-200 and ImageNet-1K.

**Full-spectrum benchmarks.** The only difference with the corresponding standard benchmarks is that we further include covariate-shifted ID samples $\mathcal{D}_{\text{csID}}^{\texttt{test}}$ and consider

$\mathcal{D}_{\text{csID}}^{\text{test}}$, $\mathcal{D}_{\text{ID}}^{\text{test}}$ together as ID. OOD data sets remain the same as in standard benchmarks so that a direct comparison can be made between the standard and full-spectrum scenario. For full-spectrum benchmarks we consider the large-scale settings of ImageNet-200 and ImageNet-1K.

We use three different $\mathcal{D}_{\text{csID}}^{\text{test}}$: ImageNet-C (Hendrycks and Dietterich, 2019) with image corruptions, ImageNet-R (Hendrycks et al., 2021a) with style changes, and ImageNet-V2 (Recht et al., 2019) with resampling bias. They are all commonly used for evaluating classifiers' generalization to covariate-shifted images. ImageNet-C has 15 corruption types, and each comes with 5 severities. We randomly sample 10K images uniformly across the 75 combinations to form the test set that is used in OpenOOD. For ImageNet-R and ImageNet-V2, we use their full data set. Note that ImageNet-C and ImageNet-V2 both have 1,000 categories (corresponding to those of ImageNet-1K), while ImageNet-R has 200 categories which are the same as those of our ImageNet-200. Therefore in the ImageNet-200 full-spectrum benchmark, we only use the 200-class subset of ImageNet-C and ImageNet-V2 as $\mathcal{D}_{\text{csID}}^{\text{test}}$.

**OOD training data.** Our benchmark also specifies OOD training samples $\mathcal{D}_{\text{OOD}}^{\text{train}}$ which can be incorporated into training when applicable (Hendrycks et al., 2019a; Yu and Aizawa, 2019; Yang et al., 2021a; Zhang et al., 2023a). To construct a meaningful $\mathcal{D}_{\text{OOD}}^{\text{train}}$ for CIFAR-10/100, we start from the 800 categories in ImageNet-1K that are apart from the 200 classes of TIN and filter out 203 categories relevant to CIFAR-10/100 based on WordNet (Miller, 1995). Named as TIN-597, the resulting data set has 597 classes which do not overlap with any of the categories from $\mathcal{D}_{\text{OOD}}^{\text{test}}$ and serves as a good candidate for $\mathcal{D}_{\text{OOD}}^{\text{train}}$ (recall that we mentioned in Section 3 why it is important to ensure $\mathcal{Y}_{\text{OOD}}^{\text{train}} \cap \mathcal{Y}_{\text{OOD}}^{\text{test}} = \varnothing$). For ImageNet-200, we directly take the rest 800 categories' images from ImageNet-1K as $\mathcal{D}_{\text{OOD}}^{\text{train}}$ (namely ImageNet-800). We do not consider $\mathcal{D}_{\text{OOD}}^{\text{train}}$ for ImageNet-1K since it is hard to find images that do not overlap with $\mathcal{D}_{\text{OOD}}^{\text{test}}$, and no relevant methods that train with OOD data have demonstrated success on ImageNet-1K.

**Comparison with prior OOD and OSR benchmarks.** Most of the OOD data sets considered in OOD detection literature fall into our far-OOD category, meaning that the more difficult near-OOD detection was less emphasized than our benchmarks do. Meanwhile, we exclude problematic OOD data sets (*e.g.*, LSUN-R and TIN-R, Liang et al., 2018, whose images contain obvious artifacts) which make detection trivial and much less meaningful (Tack et al., 2020). Following one of the seminal works (Neal et al., 2018), OSR papers often construct ID-OOD pairs by partitioning a single data set into two splits (*e.g.*, using a 6-class subset of CIFAR-10 as ID and the other 4-class subset as OOD). While such practice is well-suited for studying near-OOD detection (Ahmed and Courville, 2020), *i.e.*, ID and OOD data only has minimal covariate shifts, it inevitably reduces the scale and complexity of the problem (in terms of the number of ID and OOD categories), making the resulted benchmarks less representative for real-world scenarios. In contrast, our benchmarks hold the covariate shift between ID and near-OOD samples to a low level, while not sacrificing the scale. Specifically, all of our near-OOD groups include a certain data set that comes from the same source as ID data. For example, in the CIFAR-10 benchmark, CIFAR-100 is one of the near-OOD datsets. They are both a subset of Tiny Images (Torralba et al., 2008) and thus have minimum covariate shift between each other.

**Comparison with the benchmarks in OpenOOD v1.** Among the 6 benchmarks in v1.5, CIFAR-10/100 and ImageNet-1K standard benchmark are adapted from v1 release with necessary changes for fairness and usefulness (*e.g.*, unlike v1, v1.5 ensures that $\mathcal{D}_{\text{OOD}}^{\text{train}}$, $\mathcal{D}_{\text{OOD}}^{\text{val}}$, and $\mathcal{D}_{\text{OOD}}^{\text{test}}$ are strictly disjoint with each other). ImageNet-200 and full-spectrum benchmarks are newly introduced in v1.5. We refer readers to our changelog[3] for more details.

**Supported methods.** Like in v1 we prioritize methods that were published in top-tier machine learning conferences or journals (*e.g.*, ICML, NeurIPS, ICLR, TPAMI, etc.) and have public implementations, which can be more easily and reliably adapted into our framework. They are categorized into four groups. **Post-hoc inference methods** design post-processors, *i.e.*, the scoring function in Equation 1, that are applied to the base classifier to generate the "OOD score". They only take effect at inference phase and by default assume that the classifier is trained with the standard cross-entropy loss. In contrast, **training methods** involve training-time regularization. Most of them assume no access to auxiliary OOD training data (**w/o outlier data**), while some methods do (**w/ outlier data**). We also consider several **data augmentation** methods. An overview of each method is provided in Appendix C. Compared with v1, OpenOOD v1.5 further includes 5 more post-hoc methods, 4 more training methods, and 5 more data augmentations. Currently, OpenOOD supports **40** advanced methodologies in total for OOD detection.

## 5 Experiment Setup

We perform extensive experiments to evaluate a wide range of methods on the supported benchmarks. This section describes the training and evaluation setup of our benchmarking experiments.

**Training.** For CIFAR-10/100 and ImageNet-200, we train a ResNet-18 (He et al., 2016) for 100 epochs. We consider the standard cross-entropy training for post-hoc methods. The optimizer is SGD with a momentum of 0.9. We use a learning rate of 0.1 with cosine annealing decay schedule (Loshchilov and Hutter, 2016). A weight decay of 0.0005 is applied. The batch size is 128 for CIFAR-10/100 and 256 for ImageNet-200. Some methods have specific setup, and we adopt their official implementations and hyperparameters whenever possible.

For ImageNet-1K, we evaluate post-hoc methods with pre-trained models from torchvision (maintainers and contributors, 2016). In addition to ResNet-50 that is considered in OpenOOD v1, v1.5 further includes ViT (Dosovitskiy et al., 2021) and Swin Transformer (Liu et al., 2021) architecture for comprehensive evaluation. For training methods, we focus on ResNet-50 and use official checkpoints when possible. Otherwise, we fine-tune the torchvision pre-trained checkpoint for 30 epochs with a learning rate of 0.001. Again, we use a batch size of 256 and weight decay of 0.0005. CIFAR and ImageNet models are trained using 1 and 2 Quadro RTX 6000 GPUs (24GB memory), respectively. Except ImageNet-1K experiments, we perform 3 independent training runs for each method. Note that in the previous version of OpenOOD, the results were reported only with 1 training run.

---

3. `https://github.com/Jingkang50/OpenOOD/wiki/OpenOOD-v1.5-change-log`

We spent great efforts in maximizing reproducibility. Specifically, all training runs can be easily reproduced by running OpenOOD with configuration files.[4] We refer to our online code repo for details, which thoroughly documents all bash training scripts.[5]

**Evaluation.** As aforementioned, we use AUROC, AUPR, and FPR@95 as metrics. In the paper we focus on near-OOD and far-OOD AUROC which are averaged over all OOD data sets in each group. AUROC can be interpreted as the probability that the detector correctly separates ID and OOD samples; the random-guessing baseline is 50%, and the higher the better. Results under other metrics and per-data set statistics are available in an online table.[6] For post-hoc methods, OpenOOD supports automatic hyperparameter search using ID and OOD validation samples. The hyperparameter that yields the best AUROC is used for the final test. Similar to training, evaluation can be performed by running simple bash scripts, which again can be found in our online code repo.[7]

**Notes on missing results.** The main results for standard OOD detection are presented in Table 2. A few numbers are missing for the following reasons. OpenGAN (Kong and Ramanan, 2021) has not shown success on ImageNet-1K, and substantial changes are required to make it work with ImageNet models. CSI (Tack et al., 2020), VOS (Du et al., 2022), and NPOS (Tao et al., 2023) are infeasible with our compute resources on ImageNet. CIDER (Ming et al., 2023) and NPOS trains the CNN backbone without the final linear classifier, and the exact code for evaluating ID accuracy is not provided in their official implementations. Lastly, as aforementioned, we do not consider training with outlier data $\mathcal{D}_{\text{OOD}}^{\text{train}}$ on ImageNet-1K since it is difficult to find OOD training samples that do not overlap with test OOD data, and no relevant methods have considered ImageNet-1K.

## 6 Analysis

In this section, we discuss multiple observations and insights that arise from the benchmarking efforts of OpenOOD v1.5. Some of them are surprising while some might be less unexpected (*i.e.*, aligning with one's intuition). Nonetheless, we remark that all our observations are informative and valuable given their comprehensive nature, *i.e.*, involving 40 methods across multiple test environments. Also, to our knowledge there is no prior work that has provided similar findings to ours, especially at the scale of our work.

We start by analyzing Table 2, which presents main results of standard OOD detection on 4 benchmarks. Then we specifically look at full-spectrum detection, whose results are summarized in Figure 5. Lastly, we provide initial inspection of large foundation models (CLIP, Radford et al., 2021 and DINOv2, Oquab et al., 2023) on the task of OOD detection.

### 6.1 Standard OOD Detection

**No single winner.** In Table 2, there is no single method that consistently outperforms others across benchmarks, and the ranking can be quite different from one data set to another.

---

4. For example, `python main.py -config configs/cifar10.yml configs/resnet18_32x32.yml` will initiate a CIFAR-10 training run with ResNet-18.

5. `https://github.com/Jingkang50/OpenOOD/tree/main/scripts`

6. `https://docs.google.com/spreadsheets/d/1mTFrO-_STYBRcNMMEmHQrFPQzeg6S8Z2vRA8jawTwBw/edit?usp=sharing`

7. See Footnote 5.

| | CIFAR-10 (ResNet-18) | | | CIFAR-100 (ResNet-18) | | | ImageNet-200 (ResNet-18) | | | ImageNet-1K (ResNet-50) | | |
|---|---|---|---|---|---|---|---|---|---|---|---|---|
| | Near-OOD | Far-OOD | ID Acc. | Near-OOD | Far-OOD | ID Acc. | Near-OOD | Far-OOD | ID Acc. | Near-OOD | Far-OOD | ID Acc. |
| **- Post-hoc Inference Methods** | | | | | | | | | | | | |
| OpenMax (Bendale and Boult, 2016) | $87.62_{(\pm 0.29)}$ | $89.62_{(\pm 0.19)}$ | $95.06_{(\pm 0.30)}$ | $76.41_{(\pm 0.25)}$ | $79.48_{(\pm 0.41)}$ | $77.25_{(\pm 0.10)}$ | $80.27_{(\pm 0.10)}$ | $90.20_{(\pm 0.17)}$ | $86.37_{(\pm 0.08)}$ | 74.77 | 89.26 | 76.18 |
| MSP (Hendrycks and Gimpel, 2017) | $88.03_{(\pm 0.25)}$ | $90.73_{(\pm 0.43)}$ | $95.06_{(\pm 0.30)}$ | $80.27_{(\pm 0.11)}$ | $77.76_{(\pm 0.44)}$ | $77.25_{(\pm 0.10)}$ | $83.34_{(\pm 0.06)}$ | $90.13_{(\pm 0.09)}$ | $86.37_{(\pm 0.08)}$ | 76.02 | 85.23 | 76.18 |
| TempScale (Guo et al., 2017) | $88.09_{(\pm 0.31)}$ | $90.97_{(\pm 0.52)}$ | $95.06_{(\pm 0.30)}$ | $80.90_{(\pm 0.07)}$ | $78.74_{(\pm 0.51)}$ | $77.25_{(\pm 0.10)}$ | $\mathbf{83.69}_{(\pm 0.04)}$ | $90.82_{(\pm 0.09)}$ | $86.37_{(\pm 0.08)}$ | 77.14 | 87.56 | 76.18 |
| ODIN (Liang et al., 2018) | $82.87_{(\pm 1.85)}$ | $87.96_{(\pm 0.61)}$ | $95.06_{(\pm 0.30)}$ | $79.90_{(\pm 0.11)}$ | $79.28_{(\pm 0.21)}$ | $77.25_{(\pm 0.10)}$ | $80.27_{(\pm 0.08)}$ | $91.71_{(\pm 0.19)}$ | $86.37_{(\pm 0.08)}$ | 74.75 | 89.47 | 76.18 |
| MDS (Lee et al., 2018) | $84.20_{(\pm 2.40)}$ | $89.72_{(\pm 1.36)}$ | $95.06_{(\pm 0.30)}$ | $58.69_{(\pm 0.09)}$ | $69.39_{(\pm 1.39)}$ | $77.25_{(\pm 0.10)}$ | $61.93_{(\pm 0.51)}$ | $74.72_{(\pm 0.26)}$ | $86.37_{(\pm 0.08)}$ | 55.44 | 74.25 | 76.18 |
| MDSEns (Lee et al., 2018) | $60.43_{(\pm 0.26)}$ | $73.90_{(\pm 0.27)}$ | $95.06_{(\pm 0.30)}$ | $46.31_{(\pm 0.24)}$ | $66.00_{(\pm 0.69)}$ | $77.25_{(\pm 0.10)}$ | $54.32_{(\pm 0.24)}$ | $69.27_{(\pm 0.57)}$ | $86.37_{(\pm 0.08)}$ | 49.67 | 67.52 | 76.18 |
| RMDS (Ren et al., 2021) | $89.80_{(\pm 0.28)}$ | $92.20_{(\pm 0.21)}$ | $95.06_{(\pm 0.30)}$ | $80.15_{(\pm 0.11)}$ | $\mathbf{82.92}_{(\pm 0.42)}$ | $77.25_{(\pm 0.10)}$ | $82.57_{(\pm 0.25)}$ | $88.06_{(\pm 0.34)}$ | $86.37_{(\pm 0.08)}$ | 76.99 | 86.38 | 76.18 |
| Gram (Sastry and Oore, 2020) | $58.66_{(\pm 4.83)}$ | $71.73_{(\pm 3.20)}$ | $95.06_{(\pm 0.30)}$ | $51.66_{(\pm 0.77)}$ | $73.36_{(\pm 1.08)}$ | $77.25_{(\pm 0.10)}$ | $67.67_{(\pm 1.07)}$ | $71.19_{(\pm 0.24)}$ | $86.37_{(\pm 0.08)}$ | 61.70 | 79.71 | 76.18 |
| EBO (Liu et al., 2020) | $87.58_{(\pm 0.46)}$ | $91.21_{(\pm 0.92)}$ | $95.06_{(\pm 0.30)}$ | $80.91_{(\pm 0.08)}$ | $79.77_{(\pm 0.61)}$ | $77.25_{(\pm 0.10)}$ | $82.50_{(\pm 0.05)}$ | $90.86_{(\pm 0.21)}$ | $86.37_{(\pm 0.08)}$ | 75.89 | 89.47 | 76.18 |
| OpenGAN (Kong and Ramanan, 2021) | $53.71_{(\pm 7.68)}$ | $54.61_{(\pm 15.51)}$ | $95.06_{(\pm 0.30)}$ | $65.98_{(\pm 1.26)}$ | $67.88_{(\pm 7.16)}$ | $77.25_{(\pm 0.10)}$ | $59.79_{(\pm 3.39)}$ | $73.15_{(\pm 4.07)}$ | $86.37_{(\pm 0.08)}$ | N/A | N/A | N/A |
| GradNorm (Huang et al., 2021) | $54.90_{(\pm 0.98)}$ | $57.55_{(\pm 3.22)}$ | $95.06_{(\pm 0.30)}$ | $70.13_{(\pm 0.47)}$ | $69.14_{(\pm 1.05)}$ | $77.25_{(\pm 0.10)}$ | $72.75_{(\pm 0.48)}$ | $84.26_{(\pm 0.87)}$ | $86.37_{(\pm 0.08)}$ | 72.96 | 90.25 | 76.18 |
| ReAct (Sun et al., 2021) | $87.11_{(\pm 0.61)}$ | $90.42_{(\pm 1.41)}$ | $95.06_{(\pm 0.30)}$ | $80.77_{(\pm 0.05)}$ | $80.39_{(\pm 0.49)}$ | $77.25_{(\pm 0.10)}$ | $81.87_{(\pm 0.98)}$ | $92.31_{(\pm 0.56)}$ | $86.37_{(\pm 0.08)}$ | 77.38 | 93.67 | 76.18 |
| MLS (Hendrycks et al., 2022) | $87.52_{(\pm 0.47)}$ | $91.10_{(\pm 0.89)}$ | $95.06_{(\pm 0.30)}$ | $\mathbf{81.05}_{(\pm 0.07)}$ | $79.67_{(\pm 0.57)}$ | $77.25_{(\pm 0.10)}$ | $82.90_{(\pm 0.04)}$ | $91.11_{(\pm 0.19)}$ | $86.37_{(\pm 0.08)}$ | 76.46 | 89.57 | 76.18 |
| KLM (Hendrycks et al., 2022) | $79.19_{(\pm 0.80)}$ | $82.68_{(\pm 0.21)}$ | $95.06_{(\pm 0.30)}$ | $76.56_{(\pm 0.25)}$ | $76.24_{(\pm 0.52)}$ | $77.25_{(\pm 0.10)}$ | $80.76_{(\pm 0.08)}$ | $88.53_{(\pm 0.11)}$ | $86.37_{(\pm 0.08)}$ | 76.64 | 87.60 | 76.18 |
| VIM (Wang et al., 2022) | $88.68_{(\pm 0.28)}$ | $\mathbf{93.48}_{(\pm 0.24)}$ | $95.06_{(\pm 0.30)}$ | $74.98_{(\pm 0.13)}$ | $81.70_{(\pm 0.62)}$ | $77.25_{(\pm 0.10)}$ | $78.68_{(\pm 0.24)}$ | $91.26_{(\pm 0.19)}$ | $86.37_{(\pm 0.08)}$ | 72.08 | 92.68 | 76.18 |
| KNN (Sun et al., 2022) | $\mathbf{90.64}_{(\pm 0.20)}$ | $92.96_{(\pm 0.14)}$ | $95.06_{(\pm 0.30)}$ | $80.18_{(\pm 0.15)}$ | $82.40_{(\pm 0.17)}$ | $77.25_{(\pm 0.10)}$ | $81.57_{(\pm 0.17)}$ | $93.16_{(\pm 0.22)}$ | $86.37_{(\pm 0.08)}$ | 71.10 | 90.18 | 76.18 |
| DICE (Sun and Li, 2022) | $78.34_{(\pm 0.79)}$ | $84.23_{(\pm 1.89)}$ | $95.06_{(\pm 0.30)}$ | $79.38_{(\pm 0.23)}$ | $80.01_{(\pm 0.18)}$ | $77.25_{(\pm 0.10)}$ | $81.78_{(\pm 0.14)}$ | $90.80_{(\pm 0.31)}$ | $86.37_{(\pm 0.08)}$ | 73.07 | 90.95 | 76.18 |
| RankFeat (Song et al., 2022) | $79.46_{(\pm 2.52)}$ | $75.87_{(\pm 5.06)}$ | $95.06_{(\pm 0.30)}$ | $61.88_{(\pm 1.28)}$ | $67.10_{(\pm 1.42)}$ | $77.25_{(\pm 0.10)}$ | $56.92_{(\pm 1.59)}$ | $38.22_{(\pm 3.85)}$ | $86.37_{(\pm 0.08)}$ | 50.99 | 53.93 | 76.18 |
| ASH (Djurisic et al., 2023) | $75.27_{(\pm 1.04)}$ | $78.49_{(\pm 2.58)}$ | $95.06_{(\pm 0.30)}$ | $78.20_{(\pm 0.15)}$ | $80.58_{(\pm 0.66)}$ | $77.25_{(\pm 0.10)}$ | $82.38_{(\pm 0.19)}$ | $\mathbf{93.90}_{(\pm 0.27)}$ | $86.37_{(\pm 0.08)}$ | **78.17** | **95.74** | 76.18 |
| SHE (Zhang et al., 2023b) | $81.54_{(\pm 0.51)}$ | $85.32_{(\pm 1.43)}$ | $95.06_{(\pm 0.30)}$ | $78.95_{(\pm 0.18)}$ | $76.92_{(\pm 1.16)}$ | $77.25_{(\pm 0.10)}$ | $80.18_{(\pm 0.25)}$ | $89.81_{(\pm 0.61)}$ | $86.37_{(\pm 0.08)}$ | 73.78 | 90.92 | 76.18 |
| **- Training Methods (w/o Outlier Data)** | | | | | | | | | | | | |
| ConfBranch (DeVries and Taylor, 2018) | $89.84_{(\pm 0.24)}$ | $92.85_{(\pm 0.29)}$ | $94.88_{(\pm 0.05)}$ | $71.60_{(\pm 0.62)}$ | $68.90_{(\pm 1.83)}$ | $76.59_{(\pm 0.27)}$ | $79.10_{(\pm 0.24)}$ | $90.43_{(\pm 0.18)}$ | $85.92_{(\pm 0.07)}$ | 70.66 | 83.94 | 75.63 |
| RotPred (Hendrycks et al., 2019b) | $\mathbf{92.68}_{(\pm 0.27)}$ | $96.62_{(\pm 0.18)}$ | $\mathbf{95.35}_{(\pm 0.52)}$ | $76.43_{(\pm 0.16)}$ | $\mathbf{88.40}_{(\pm 0.13)}$ | $76.03_{(\pm 0.38)}$ | $81.59_{(\pm 0.20)}$ | $92.56_{(\pm 0.09)}$ | $\mathbf{86.37}_{(\pm 0.16)}$ | **76.52** | 90.00 | **76.55** |
| G-ODIN (Hsu et al., 2020) | $89.12_{(\pm 0.57)}$ | $95.51_{(\pm 0.31)}$ | $94.70_{(\pm 0.25)}$ | $77.15_{(\pm 0.28)}$ | $85.67_{(\pm 1.58)}$ | $74.46_{(\pm 0.04)}$ | $77.28_{(\pm 0.10)}$ | $92.33_{(\pm 0.11)}$ | $84.56_{(\pm 0.28)}$ | 70.77 | 85.51 | 74.85 |
| CSI (Tack et al., 2020) | $89.51_{(\pm 0.19)}$ | $92.00_{(\pm 0.30)}$ | $91.16_{(\pm 0.14)}$ | $71.45_{(\pm 0.27)}$ | $66.31_{(\pm 1.21)}$ | $61.60_{(\pm 0.46)}$ | N/A | N/A | N/A | N/A | N/A | N/A |
| ARPL (Chen et al., 2021) | $87.44_{(\pm 0.15)}$ | $89.31_{(\pm 0.32)}$ | $93.66_{(\pm 0.11)}$ | $74.94_{(\pm 0.93)}$ | $73.69_{(\pm 1.80)}$ | $70.70_{(\pm 1.08)}$ | $82.02_{(\pm 0.10)}$ | $89.23_{(\pm 0.11)}$ | $83.95_{(\pm 0.32)}$ | 76.30 | 85.50 | 75.87 |
| MOS (Huang and Li, 2021) | $71.45_{(\pm 3.09)}$ | $76.41_{(\pm 5.93)}$ | $94.83_{(\pm 0.37)}$ | $80.40_{(\pm 0.18)}$ | $80.17_{(\pm 1.21)}$ | $76.98_{(\pm 0.20)}$ | $69.84_{(\pm 0.46)}$ | $80.46_{(\pm 0.92)}$ | $85.60_{(\pm 0.20)}$ | 72.85 | 82.75 | 72.81 |
| VOS (Du et al., 2022) | $87.70_{(\pm 0.48)}$ | $90.83_{(\pm 0.92)}$ | $94.31_{(\pm 0.64)}$ | $\mathbf{80.93}_{(\pm 0.29)}$ | $81.32_{(\pm 0.09)}$ | $77.20_{(\pm 0.20)}$ | $82.51_{(\pm 0.11)}$ | $91.00_{(\pm 0.28)}$ | $86.23_{(\pm 0.19)}$ | N/A | N/A | N/A |
| LogitNorm (Wei et al., 2022) | $92.33_{(\pm 0.08)}$ | $\mathbf{96.74}_{(\pm 0.06)}$ | $94.30_{(\pm 0.25)}$ | $78.47_{(\pm 0.31)}$ | $81.53_{(\pm 1.26)}$ | $76.34_{(\pm 0.17)}$ | $\mathbf{82.66}_{(\pm 0.15)}$ | $93.04_{(\pm 0.21)}$ | $86.04_{(\pm 0.15)}$ | 74.62 | 91.54 | 76.45 |
| CIDER (Ming et al., 2023) | $90.71_{(\pm 0.16)}$ | $94.71_{(\pm 0.36)}$ | N/A | $73.10_{(\pm 0.39)}$ | $80.49_{(\pm 0.68)}$ | N/A | $80.58_{(\pm 1.75)}$ | $90.66_{(\pm 1.68)}$ | N/A | 68.97 | **92.18** | N/A |
| NPOS (Tao et al., 2023) | $89.78_{(\pm 0.33)}$ | $94.07_{(\pm 0.49)}$ | N/A | $78.35_{(\pm 0.37)}$ | $82.29_{(\pm 1.55)}$ | N/A | $79.40_{(\pm 0.39)}$ | $\mathbf{94.49}_{(\pm 0.07)}$ | N/A | N/A | N/A | N/A |
| **- Training Methods (w/ Outlier Data)** | | | | | | | | | | | | |
| OE (Hendrycks et al., 2019a) | $\mathbf{94.82}_{(\pm 0.21)}$ | $\mathbf{96.00}_{(\pm 0.13)}$ | $94.63_{(\pm 0.26)}$ | $\mathbf{88.30}_{(\pm 0.10)}$ | $\mathbf{81.41}_{(\pm 1.49)}$ | $\mathbf{76.84}_{(\pm 0.42)}$ | $\mathbf{84.84}_{(\pm 0.16)}$ | $\mathbf{89.02}_{(\pm 0.18)}$ | $85.82_{(\pm 0.21)}$ | N/A | N/A | N/A |
| MCD (Yu and Aizawa, 2019) | $91.03_{(\pm 0.12)}$ | $91.00_{(\pm 1.10)}$ | $94.95_{(\pm 0.04)}$ | $77.07_{(\pm 0.32)}$ | $74.72_{(\pm 0.78)}$ | $75.83_{(\pm 0.04)}$ | $83.62_{(\pm 0.09)}$ | $88.94_{(\pm 0.10)}$ | $\mathbf{86.12}_{(\pm 0.17)}$ | N/A | N/A | N/A |
| UDG (Yang et al., 2021a) | $89.91_{(\pm 0.25)}$ | $94.06_{(\pm 0.90)}$ | $92.36_{(\pm 0.84)}$ | $78.02_{(\pm 0.10)}$ | $79.59_{(\pm 1.77)}$ | $71.54_{(\pm 0.64)}$ | $74.30_{(\pm 1.63)}$ | $82.09_{(\pm 2.78)}$ | $68.11_{(\pm 1.24)}$ | N/A | N/A | N/A |
| MixOE (Zhang et al., 2023a) | $88.73_{(\pm 0.82)}$ | $91.93_{(\pm 0.69)}$ | $94.55_{(\pm 0.32)}$ | $80.95_{(\pm 0.20)}$ | $76.40_{(\pm 1.44)}$ | $75.13_{(\pm 0.06)}$ | $82.62_{(\pm 0.03)}$ | $88.27_{(\pm 0.41)}$ | $85.71_{(\pm 0.07)}$ | N/A | N/A | N/A |

Table 2: Main results from OpenOOD v1.5 on standard OOD detection. In this table we use AUROC as the metric for OOD detection. Whenever applicable, we report the average number and the corresponding standard deviation obtained from 3 training runs. The best result within each group is bolded. Results for data augmentation methods are listed in Table 3. Full result table including other metrics and per-dataset statistics can be found in an online sheet (see Footnote 6).

For example, ReAct (Sun et al., 2021) and ASH (Djurisic et al., 2023) are extremely powerful on ImageNet but less competitive on CIFAR. In contrast, methods that yield remarkable performance on small data sets (*e.g.*, KNN, Sun et al., 2022 and RotPred, Hendrycks et al., 2019b) do not show clear advantage on large data sets. We believe the absence of a clear winner can be attributed, in part, to the evaluation inconsistencies of current methods, underscoring the importance of our benchmark.

**Data augmentations help.** While data augmentations have been shown beneficial for standard classification (Cubuk et al., 2020) and OOD generalization (Geirhos et al., 2019; Hendrycks et al., 2020, 2021a,c; Pinto et al., 2022), their effects for OOD detection remain unclear. In Table 3, we find that several data augmentation methods, despite not being designed to improve OOD detection, can actually boost detection rates in many cases. More interestingly, the performance gain is amplified when they are combined with powerful

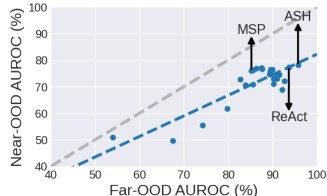

Figure 2: Near-OOD improvements are proportional to, yet slower than, far-OOD improvements on ImageNet-1K.

| | ImageNet-200 | | | ImageNet-1K | | | |
|---|---|---|---|---|---|---|---|
| | MSP | ASH | ID Acc. | MSP | ReAct | ASH | ID Acc. |
| CrossEntropy | 83.34 / 90.13 | 82.38 / 93.90 | 86.37 | 76.02 / 85.23 | 77.38 / 93.67 | 78.17 / 95.74 | 76.18 |
| StyleAugment (Geirhos et al., 2019) | 80.99 / 88.44 | 80.65 / 93.70 | 83.41 | 75.78 / 85.73 | 76.70 / 91.88 | 78.21 / 94.90 | 74.68 |
| RandAugment (Cubuk et al., 2020) | 83.17 / 90.34 | 81.56 / 94.53 | 86.58 | 76.60 / 85.27 | 78.30 / 93.50 | 79.81 / 95.01 | 76.90 |
| AugMix (Hendrycks et al., 2020) | 83.49 / 90.68 | **82.87** / 94.66 | 87.01 | **77.49** / **86.67** | **79.94** / **93.70** | **82.16** / **96.05** | **77.63** |
| DeepAugment (Hendrycks et al., 2021a) | 81.39 / 88.79 | 80.61 / 93.84 | 85.00 | 76.67 / 86.26 | 78.43 / 92.12 | 79.14 / 93.90 | 76.77 |
| PixMix (Hendrycks et al., 2021c) | 82.15 / 90.23 | 81.36 / **95.01** | 85.79 | 76.86 / 85.63 | 79.12 / 91.59 | 78.92 / 92.17 | 77.44 |
| RegMixup (Pinto et al., 2022) | **84.13** / **90.81** | 79.38 / 92.74 | **87.25** | 77.04 / 86.31 | 77.68 / 92.45 | 78.45 / 95.35 | 76.68 |

Table 3: Data augmentation methods (column headers) are beneficial for OOD detection and amplify the performance gain when combined with post-hoc methods (row headers). The cell numbers represent the near-OOD / far-OOD AUROC.

post-processors. For example, compared with the baseline of cross-entropy training on ImageNet-1K near-OOD, AugMix (Hendrycks et al., 2020) achieves 76.02 + **1.47** = 77.49% AUROC and 78.17 + **3.99** = 82.16% AUROC when working with MSP (Hendrycks and Gimpel, 2017) and ASH (Djurisic et al., 2023), respectively. 82.16% is the current best score among all methods. The results indicate that the effects from data augmentations and post-processors are complementary.

As to why data augmentations benefit OOD detection, one potential explanation is that having more diverse training samples can help models better capture semantic-correlated features rather than spurious features. Spurious features—a type of features that correlate to the label but are not semantically meaningful—are pervasive in natural images (examples include high-frequency patterns, Gilmer and Hendrycks, 2019 and adversarial noises, Ilyas et al., 2019). Models that heavily learn such spurious features can be easily activated by certain spurious features presented in OOD samples (Ming et al., 2022b). Data augmentations, in contrast, introduce diverse semantic-preserving training samples which can guide the model to focus more on features that semantically correlate with the ID categories. As a result, at inference time, models trained with data augmentations may only respond to semantic-meaningful features of ID samples and activate less in the face of OOD samples, making ID and OOD data more separable.

**Near-OOD remains more challenging than far-OOD.** Figure 2 plots the trend of near-OOD AUROC v.s far-OOD AUROC on ImageNet-1K. Not surprisingly, near-OOD AUROC is (roughly) proportional to far-OOD AUROC, meaning that the improvement for one group is likely to help with the other as well. Meanwhile, we notice that the progress on near-OOD is slower than that on far-OOD. Besides the fact that near-OOD detection is more difficult, this may also be due to that previous works mainly focus on far-OOD data sets when designing and evaluating their methods.

**Vision transformers do not outperform ResNets.** We visualize in Figure 3 the performance of a few powerful post-hoc methods on ImageNet-1K with ResNet-50, ViT-B-16 (Dosovitskiy et al., 2021), and Swin-T (Liu et al., 2021) being the classifier. We use the ImageNet-1K pre-trained checkpoints provided by torchvision. We find that vi-

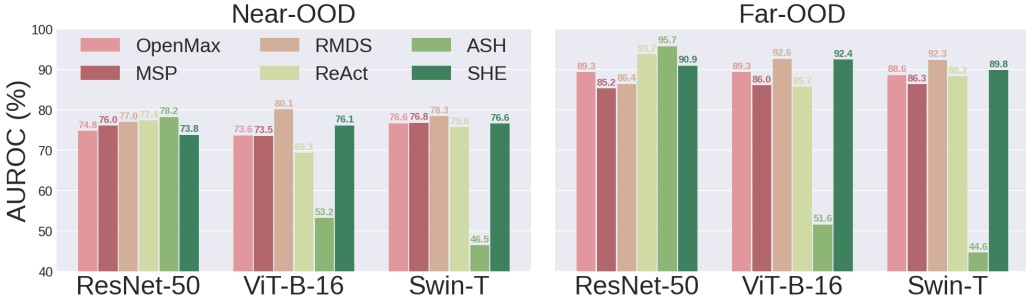

Figure 3: OOD detection rates of post-hoc methods with different architectures on ImageNet-1K. Some methods are sensitive to model architecture while some are not. Transformers do not seem to have clear advantage over ResNets.

sion transformers do *not* show noticeable improvements over ResNets for OOD detection. Meanwhile, different post-processor may favor different architecture. For instance, the top-2 post-processor on ResNet-50, *i.e.*, ASH (Djurisic et al., 2023) and ReAct (Sun et al., 2021), both have significant performance degradation when operating on transformer. RMDS (Ren et al., 2021), in contrast, suits transformers much better than ResNets.

Note the ID accuracy of the considered ResNet-50, ViT-B, and Swin-T is 76.18%, 81.14%, and 81.59%, respectively. Since OOD detection performance often correlates with ID accuracy (Vaze et al., 2022), we initially expect that transformers would yield superior OOD detection capability as well. Based on our above finding that post-processors could be sensitive to model architecture, we suspect that the reason for not seeing substantial advantages in transformers is that current post-processors do not suit them well: Indeed, most post-hoc methods are tailored to CNN architectures such as ResNets.

**Training methods excel at small data sets.** On CIFAR-10/100, we find that training-time regularizations (the second group in Table 2) can provide better OOD detection capability than post-hoc methods. In particular, RotPred (Hendrycks et al., 2019b) and LogitNorm (Wei et al., 2022) stand out as two powerful training methods (without using outlier data). On CIFAR-10, they both lead to ∼2% and ∼3% increase in near- and far-OOD AUROC, respectively, compared to the best-performing post-processors. Meanwhile, however, training methods in general do not outperform post-hoc ones on ImageNet-200/1K. This might be due to that more sophisticated training dynamics require larger models or longer training.

**Post-hoc methods are more effective for large-scale settings.** This is particularly true on ImageNet-1K, where applying post-processors to a model pre-trained with the standard cross-entropy loss are the top-performing solutions for both near- and far-OOD detection, according to Table 2.

**Outlier data helps in certain cases.** Compared with methods that do not have such consideration, incorporating OOD training data (the last group in Table 2) is helpful mainly when the test OOD samples are similar to the training ones. For instance, OE (Hendrycks et al., 2019a) yields the highest near-OOD AUROC on CIFAR-100 because the used OOD training set (TIN-597; see Section 4 for details) is similar to one of the near-OOD test sets

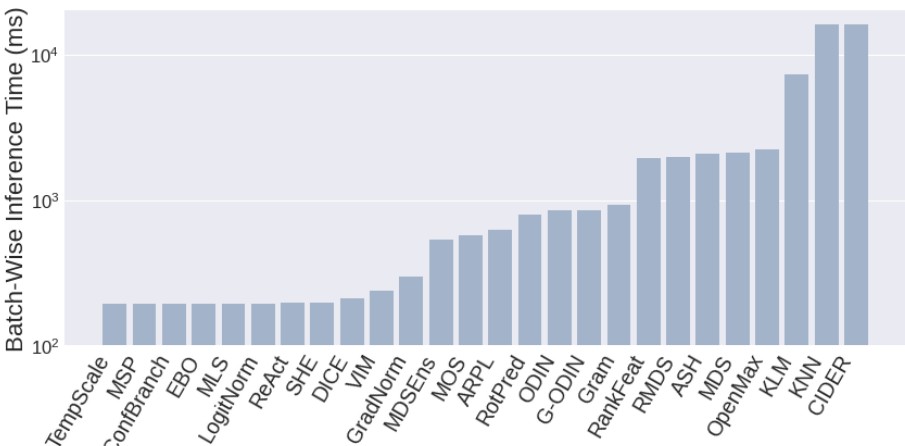

Figure 4: Inference time of each method (in milliseconds and sorted from left to right) on a batch of 200 ImageNet 224x224 images. The base model is ResNet-50. The inference time is profiled with a single 24GB GPU, and we report the average results over 5 runs. We notice that some methods incur significantly larger inference cost than others.

(TIN). In contrast, it does not seem to be beneficial for detecting far-OOD samples (which are quite different from TIN images) and actually underperforms several other methods which do not use the outlier data. An interesting future direction would be to study whether outliers that are less close to the actual test OOD distribution can still contribute and benefit the detector. Such investigation is particularly meaningful in fine-grained applications (Zhang et al., 2023a) or data-scarce scenarios.

**ID accuracy can be affected a little.** Lastly, as shown in Table 2, most training methods incur slight drop (within 1%) in ID classification accuracy compared to the standard cross-entropy training. For some methods the drop could be large, and it is important for future works to monitor ID accuracy to maintain utility while improving OOD detection capability.

**The time and space cost.** We show in Figure 4 the batch-wise inference time cost of OOD detection methods on ImageNet with ResNet-50 as the base model. Specifically, we choose 200 as a reasonably large batch size such that for all methods the forward pass won't raise Out-of-Memory error on a 24GB GPU. We count as the start of inference when the model receives the batched inputs and as the end when OOD scores are output from the detector, excluding the time cost resulted from other activities such as data loading. The reported numbers are averaged over 5 runs (with an extra dry run in the beginning). From Figure 4, we see that several methods have low time complexity since they introduce small to little change to the standard forward pass of the neural network (*e.g.*, MSP, MLS, and ReAct). However, many other methods that involve sophisticated design of computing the OOD score (*e.g.*, RotPred, ASH, KNN, CIDER) could incur orders of magnitude increase in the inference cost, which may significantly limit their use in practice.

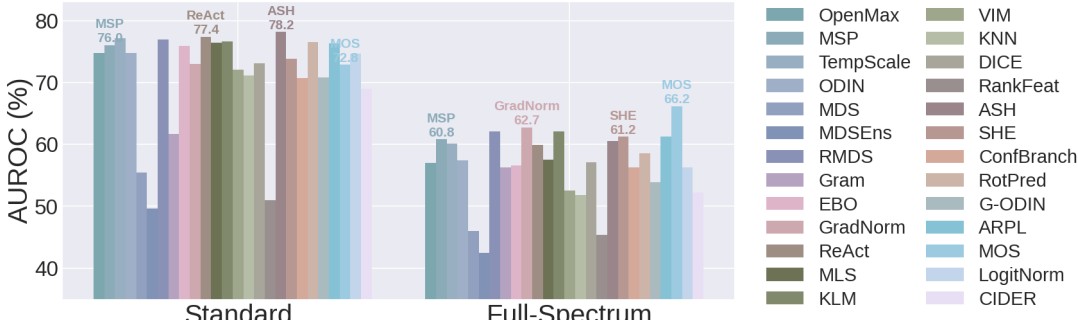

Figure 5: Comparison between standard and full-spectrum detection on ImageNet-1K (near-OOD). Many detectors suffer significant performance degradation in the full-spectrum setting.

For space cost, we give a conceptual analysis that considers the auxiliary cost that each OOD detection method introduce. Similar to time cost, several methods such as MSP and MLS have little to no extra space cost. Meanwhile, a few methods such as MDS, KNN, KLM, and CIDER require the feature vectors of ID data to compute OOD score, thus inducing great space complexity especially when the number of ID feature vectors stored in the memory is large. To our knowledge, there is very few OOD detection work that takes space cost (and/or time cost) into consideration. Our results and analysis here point out the need for future works to account for these aspects and explore the performance-complexity trade-off when designing new methods.

## 6.2 Full-Spectrum Detection

**Full-spectrum detection poses challenge for current detectors.** This can be seen in Figure 5, where the near-OOD AUROC of most methods decreases by >10% on ImageNet-1K (similar trend holds for far-OOD; see full results via the link in Footnote 6). The performance drop suggests that *existing OOD detectors can be sensitive to the non-semantic covariate shift and are likely to flag covariate-shifted ID samples as OOD.*

Such behaviour is not ideal because: 1) It does not align with human perception/decision (*e.g.*, a human annotator classifying `dog` and `car` wouldn't mark a covariate-shifted dog image as something unknown or novel). 2) It can harm the classifier's generalization capability on covariate-shifted ID data, as in practice it is often assumed that the classifier would refrain from making predictions when the sample is identified as OOD (or at least the OOD flag associated with the prediction would indicate its unreliability). As a result, we firmly believe that full-spectrum detection is an important open problem for ensuring human-model alignment and the model's practical reliability. One method that stands out in this resort is MOS (Huang and Li, 2021), which utilizes a two-level semantic hierarchy to facilitate classification and OOD detection. MOS exhibits the smallest performance drop when changing from standard to full-spectrum detection (from 72.85% to 66.17% in AUROC) and serves as a strong baseline for future works to tackle full-spectrum detection.

|  | MSP | GradNorm | SHE | ID Acc. |
|---|---|---|---|---|
| CrossEntropy | 60.79 / 72.32 | 62.70 / **83.49** | 61.21 / 83.04 | 54.35 |
| StyleAugment (Geirhos et al., 2019) | 62.09 / 74.37 | 65.27 / 81.62 | 66.64 / 82.64 | 55.44 |
| RandAugment (Cubuk et al., 2020) | 61.36 / 72.07 | 63.27 / 76.08 | 64.41 / 76.68 | 55.57 |
| AugMix (Hendrycks et al., 2020) | 63.14 / 74.62 | **67.10** / 81.29 | **69.66** / **83.06** | 57.46 |
| DeepAugment (Hendrycks et al., 2021a) | 63.51 / **75.40** | 65.66 / 76.27 | 68.27 / 78.85 | **57.82** |
| PixMix (Hendrycks et al., 2021c) | **62.51** / 73.47 | 61.07 / 70.00 | 65.02 / 77.03 | 57.27 |
| RegMixup (Pinto et al., 2022) | 61.32 / 72.87 | 61.86 / 79.98 | 64.71 / 81.23 | 55.55 |

Table 4: ImageNet-1K full-spectrum detection results of data augmentation methods. The numbers in each cell represent the near-OOD / far-OOD AUROC. Again, data augmentations are helpful especially when combined with approriate post-processors and when performing near-OOD detection.

**Data augmentations continue to help in full-spectrum settings.** Similar to our observations in standard OOD detection, in Table 4 we see that data augmentations also boost full-spectrum detection rates especially when combined with powerful post-hoc methods. For example, compared to the cross-entropy baseline, while AugMix (Hendrycks et al., 2020) increases near-OOD AUROC "only" by 2.35% with the MSP detector (Hendrycks and Gimpel, 2017), it leads to a much significant improvement of 8.45% when working with SHE (Zhang et al., 2023b). AugMix + SHE is the current best approach in terms of full-spectrum near-OOD AUROC on ImageNet-1K. In the meantime, we do notice that data augmentations do not clearly benefits full-spectrum far-OOD AUROC, and the reasons require future study. That said, in general our results demonstrate that "data augmentation + post-processor" is promising for both standard and full-spectrum OOD detection.

Finally, we note that several data augmentations evaluated in Table 4 are in fact established methods in OOD generalization research. Do other OOD generalization methods help with OOD detection, especially in full-spectrum settings? This is one particular interesting question that we encourage future works to explore, because if the answer is "yes", then it would be promising for researchers in OOD generalization and OOD detection community to collaborate and improve the model's robustness against both covariate and semantic distribution shifts together.

## 6.3 Foundation Models

As another effort in extending the scope of OpenOOD and connecting the field with recent advances, in this section we turn our attention to foundation models. Foundation models are large pre-trained models that can be adapted for a wide range of tasks; they have demonstrated superior recognition capability over the task-specific, strictly-supervised counterparts (Radford et al., 2021). In particular, zero-shot foundation models generalize significantly better when facing *covariate*-shifted samples when it comes to OOD generalization (Wortsman et al., 2022). This motivates us to see whether the same advantage exists when foundation models handle *semantic*-shifted samples, *i.e.*, in the context of OOD detection.

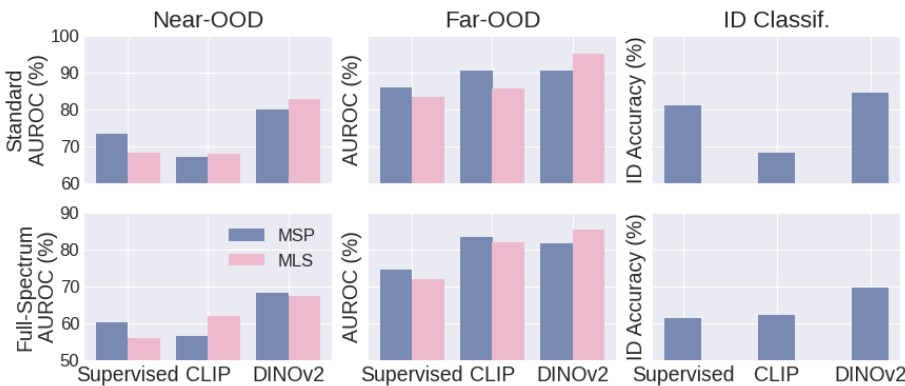

Figure 6: Performance of foundation models on ImageNet-1K. Here we consider OOD detectors that are compatible with the zero-shot CLIP (MSP and MLS).

To this end, we evaluate two foundation models, CLIP (Radford et al., 2021) and DINOv2 (Oquab et al., 2023), on our ImageNet-1K benchmark and compare them with a ImageNet-1K trained classifier. We consider zero-shot classification for CLIP (using the prompt ensemble to obtain the classifier weight as recommended[8]) given its strong generalization ability, and use linear probe[9] for DINOv2 which does not have zero-shot capability. We do not consider fine-tuning those foundation models, whose effect on OOD detection has been studied by Ming and Li (2023). All three classifiers share the same ViT-Base backbone.

The results are summarized in Figure 6. Compared with the task-specific supervised model, zero-shot CLIP shows substantial improvements on far-OOD detection in both standard and full-spectrum setting, yet the comparison on near-OOD detection is nuanced and gives no conclusive remark. However, we feel like giving any conclusion at this moment will be too early: Most existing detectors are *not* compatible with zero-shot CLIP, and the only detector that specifically targets CLIP (Ming et al., 2022a) adopts a simple design that is equivalent to the most basic detector MSP (Hendrycks and Gimpel, 2017). We hypothesize that CLIP's power may not be fully unleashed until more advanced or appropriate detector is developed. Meanwhile, we observe that the linear probe of self-supervised DINOv2 consistently and remarkably outperforms the fully-supervised counterpart, in *all* OOD detection cases and in ID classification. This demonstrates that in addition to their superior performance in closed-set classification, large-scale self-supervised foundation models are also powerful for handling semantic distribution shifts in an open world. In all, our inspection suggests that foundation models are promising directions to explore for OOD detection.

## 7 Conclusion and Discussion

In this work we present OpenOOD v1.5, which enhances its earlier version by 1) constructing more rigorous evaluation protocol and benchmarks, 2) providing comprehensive results

---

8. `https://github.com/openai/CLIP/blob/main/notebooks/Prompt_Engineering_for_ImageNet.ipynb`
9. `https://github.com/facebookresearch/dinov2#pretrained-heads---image-classification`

on the large-scale ImageNet data set, and 3) investigating full-spectrum detection. We provide several observations from our results to identify open problems and provide future directions, including but not limited to: 1) a single winner that performs competitively across multiple benchmarks is still missing; 2) the combination of data augmentations and strong post-processors are particularly effective; 3) full-spectrum detection is a challenging problem that might need insights from both OOD generalization and detection community; 4) more advanced architectures like vision transformers and models like CLIP may require more dedicated detectors to release their potential. We hope that the codebase, benchmark, evaluation results, and insights of OpenOOD can accelerate the progress and foster collective efforts towards advancing the state-of-the-art in OOD detection.

**Related work.** To the best of our knowledge, OpenOOD v1.5 is the only work that comprehensively evaluates a wide range of OOD detection methods on multiple benchmarks of various sizes. We give a detailed discussion on a few works that relate to OpenOOD in certain aspects in Appendix E.

**Limitation.** In this work we focus on the context where there assumes to be a discriminative classifier for the ID classification in the first place. As a result, all the OOD detection methods considered in our work are discriminative. OOD detection approaches that are based on generative modeling (*e.g.*, Zisselman and Tamar, 2020; Kirichenko et al., 2020; Nalisnick et al., 2018; Serrà et al., 2020) are not currently included in OpenOOD. To our knowledge, generative methods have not yet demonstrated scalability on ImageNet-level data and in general are less competitive than discriminative methods. Additional efforts will be required to integrate generative methods into OpenOOD in the future, as they often rely on dedicated/specialized model architecture and training procedure (Zisselman and Tamar, 2020; Kirichenko et al., 2020; Serrà et al., 2020).

**Real-world implications.** OOD detection is the building component in many real-world cases such as remote sensing applications (Inkawhich et al., 2022) and fine-grained novel category discovery (Zhang et al., 2023a). It also has strong implications for safety-critical applications, as OOD detection can be used to detect unexpected anomalies, unknown unknowns, and Black Swans (Hendrycks et al., 2021b). We believe that OpenOOD has laid a solid foundation for tackling OOD detection in those specific scenarios.

**Broader impact.** OOD detection is an important topic for machine learning safety as it studies how deep neural networks can handle unknown inputs desirably. As an open-sourced, unified, and comprehensive benchmark for OOD detection, OpenOOD is expected to benefit the whole community and facilitate relevant research, which we believe has positive broader impacts.

On the other hand, while OpenOOD itself as a benchmark platform does not incur concerns, certain methods that are included in OpenOOD may pose privacy or security risks. Specifically, methods such as KNN and SHE rely on the extracted representation of training samples to compute the OOD score, making it vulnerable to privacy attacks (Zhang et al., 2022). Meanwhile, Inkawhich et al. (2023) showed that OOD detectors enlarge the attack surface of deep learning systems, and existing methods can easily be compromised by adversarial attacks. Nonetheless, we believe that OpenOOD provides a good starting point to study mitigations of such negative impacts and we encourage future works to take this into consideration.

**Future work.** First, we will keep maintaining OpenOOD's codebase and leaderboard. The codebase is hosted on Github, and the leaderboard is hosted using Github pages, which both are free services. We anticipate the benchmark to be community-driven: Reporting new results and submitting new entries to the leaderboard would be easy with our unified evaluator.

In addition to the maintenance, in the future v2 release we plan to further expand the scope of OpenOOD beyond image classification and include more application scenarios such as object detection, semantic segmentation, and natural language processing tasks. Specifically, it would be interesting to see whether current OOD detectors, which are designed for image classifiers, can generalize to different problems and modalities.

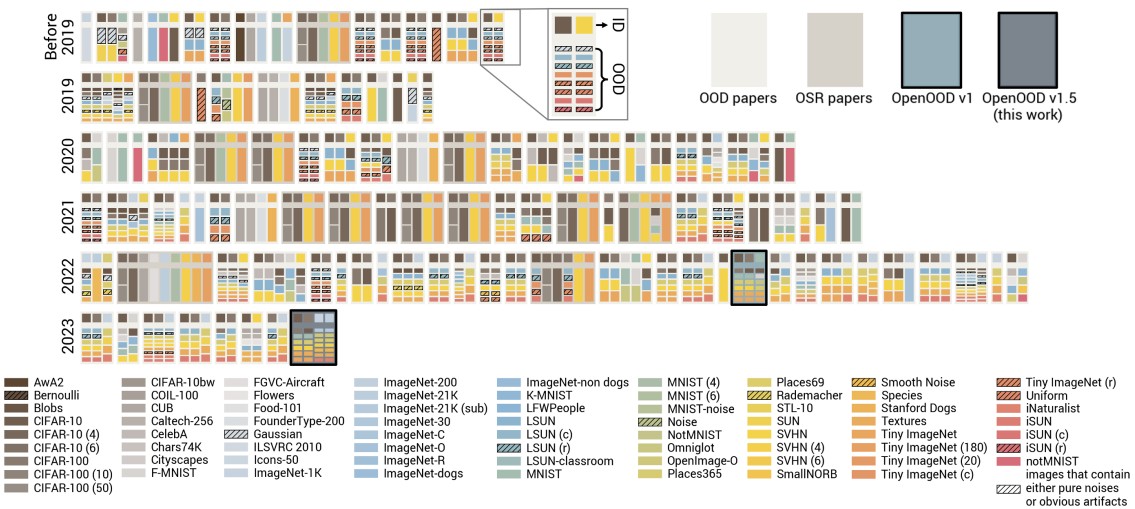

Figure 7: Summarizing evaluation settings of 100+ recent OOD detection and OSR works from NeurIPS, AAAI, ICLR, CVPR, ICML, and ICCV/ECCV (zoom in to view better). Each box stands for a paper. Within the box, each column shows the ID data set and corresponding OOD data sets which are represented by the color blocks. The lack of a consistent color pattern between boxes signifies the inconsistency in the evaluation setup of current works. Multiple works also adopted unrealistic or problematic data (Tack et al., 2020) in evaluation (marked by the hatch pattern). The community still suffers from such chaos after OpenOOD v1 (Yang et al., 2022a) came out. Full paper list used to produce this figure can be found at https://drive.google.com/file/d/1tO2uxhRJQdw-R95wJQTsqg_xsw1oHhWm/view?usp=sharing.

## Acknowledgments and Disclosure of Funding

This work is supported by National Science Foundation (NSF) CNS-2112562, NSF CNS-1822085, and ARO W911NF-23-2-0224. This study is also supported by the Ministry of Education, Singapore, under its MOE AcRF Tier 2 (MOE-T2EP20221- 0012), NTU NAP, and under the RIE2020 Industry Alignment Fund – Industry Collaboration Projects (IAF-ICP) Funding Initiative, as well as cash and in-kind contribution from the industry partner(s). Yixuan Li gratefully acknowledges the funding support by the AFOSR Young Investigator Program under award number FA9550-23-1-0184, National Science Foundation (NSF) Award No. IIS-2237037 & IIS-2331669, Office of Naval Research under grant number N00014-23-1-2643, Philanthropic Fund from SFF, and faculty research awards/gifts from Google and Meta.

## Appendix A. Evaluation Pitfalls of OOD Detection

Here we first explain the three evaluation pitfalls that we identify in the current OOD detection research.

Figure 8: **Left:** An example of evaluting ImageNet-1K models in a few lines with our `Evaluator`. **Right:** Screenshot of top entries on our ImageNet-1K leaderboard hosted at `https://zjysteven.github.io/OpenOOD/`. Zoom in to view better.

**Confusing terminologies.** Despite subtle differences in the way of constructing their test environments, OOD detection and Open-Set Recognition (OSR) (or sometimes, "novelty detection") are essentially pursuing the same goal (Bendale and Boult, 2016; Hendrycks and Gimpel, 2017). With two different terminologies, however, the two topics often diverge from each other in a counterproductive way, where methods are developed and compared separately within each branch using different benchmarks.

**Inconsistent data sets.** Given an ID data set, the simplest practice is to use other data sets with semantically different visual categories as OOD data sets. Unfortunately, we have seen great inconsistency in the selected data sets for OOD detection evaluation. Such phenomenon is highlighted in Figure 7, where we summarize the evaluation settings of 100+ recent works from top-tier machine learning conferences. The lack of a consistent pattern indicates how different the used data sets are from paper to paper, causing great difficulty for straight comparison between methods. The evaluation settings within the OSR branch are more consistent yet are significantly limited in scale (see more discussion in Section 4).

**Erroneous practices** could compromise evaluation if no extra care is taken. One example that is pervasive in OOD detection works is leaking information about OOD data that is used for evaluation. More specifically, some methods train the model or tune hyperparameters with test OOD data (Perera and Patel, 2019; Liang et al., 2018; Kong and Ramanan, 2021). Such practices go against basic machine learning principles and will lead to overoptimistic results.

## Appendix B. New Features and Updates of OpenOOD v1.5

OpenOOD v1.5 introduces new features including a *leaderboard* hosted online to track the-state-of-the-art based on various methods and a lightweight *evaluator* which enables easy evaluation with a few lines of code (see Figure 8). Other updates such as adding newer methods and fixing implementation bugs are documented in the changelog.[10]

---

10. `https://github.com/Jingkang50/OpenOOD/wiki/OpenOOD-v1.5-change-log`

## Appendix C. Supported Methods

We now overview the supported methods of OpenOOD v1.5.

**Post-Hoc Inference Methods**. Recall from Equation 1 that given an input image, OOD detectors function by assigning an "OOD score" that is computed based on certain outputs from the base classifier, which will then be thresholded to give the binary prediction. The first line of works focus on designing such post-processors/scoring mechanisms that best separate ID and OOD samples. **MSP** (Hendrycks and Gimpel, 2017) takes the maximum softmax probability over ID categories as the score. **OpenMax** (Bendale and Boult, 2016) is a replacement for the softmax layer which directly estimates the probability of an input being from an unknown class. **TempScale** (Guo et al., 2017) calibrates softmax probabilities with temperature scaling. **ODIN** (Liang et al., 2018) further introduces input preprocessing on top of TempScale. **MDS** (Lee et al., 2018) fits class-conditional Gaussian distribution on the penultimate layer features of the classifier and derives OOD score with Mahalanobis distance. **MDSEns** (Lee et al., 2018) is another version of MDS which leverages multiple intermediate layers and forms a feature ensemble. **RMDS** (Ren et al., 2021) improves MDS by considering the "background score" computed from an unconditional Gaussian distribution. **Gram** (Sastry and Oore, 2020) identifies abnormal patterns from the Gram Matrices of intermediate feature maps. **EBO** (Liu et al., 2020) applies energy function to the logits to compute OOD score. **OpenGAN** (Kong and Ramanan, 2021) trains a GAN in the classifier's feature space and uses the discriminator as the post-processor. **GradNorm** (Huang et al., 2021) computes the KL divergence between the softmax probability distribution and the uniform distribution and takes the gradients of penultimate layer weights w.r.t. the KL divergence as OOD score. **ReAct** (Sun et al., 2021) rectifies feature vectors by thresholding their elements with a certain magnitude. **MLS** (Hendrycks et al., 2022) uses the maximum logit. **KLM** (Hendrycks et al., 2022) looks at the KL divergence between the softmax probability distribution and a "template" distribution. **VIM** (Wang et al., 2022) augments the logits with the norm of feature residual compared with ID training samples' features to compute the OOD score. **KNN** (Sun et al., 2022) applies KNN to the penultimate layer's features. **DICE** (Sun and Li, 2022) sparsifies the last linear layer before computing the logits. **RankFeat** (Song et al., 2022) transforms the feature matrices such that their rank is 1. **ASH** (Djurisic et al., 2023) shapes later layer activations by removing a large portion of the elements and simplifing the rest. **SHE** (Zhang et al., 2023b) maintains a template representation for each ID category and detects OOD samples by measuring the distance between the representation of an input to that template.

**Training methods without outlier data.** Unlike post-hoc methods that only interfere with the inference process, training methods involve training-time regularization to enhance OOD detection capability. **ConfBranch** (DeVries and Taylor, 2018) trains another branch in addition to the classification one to explicitly learn the estimate of model uncertainty. **RotPred** (Hendrycks et al., 2019b) includes an extra head to predict the rotation angle of rotated inputs in a self-supervised manner, and the rotation head together with the classification head is used for OOD detection. **G-ODIN** (Hsu et al., 2020) utilizes a dividend/divisor structure and decomposes the softmax confidence for better ID-OOD separation. **CSI** (Tack et al., 2020) explores self-supervised contrastive learning objectives for OOD detectors. **ARPL** (Chen et al., 2021) introduces "reciprocal points" for each ID category

and trains the model by pushing the reciprocal point away from the corresponding ID cluster and encouraging OOD samples to gather around the reciprocal point. **MOS** (Huang and Li, 2021) incorporates a two-level hierarchical classifier and designs an accompanying OOD score to benefit OOD detection especially in large-scale settings. **VOS** (Du et al., 2022) regularizes the feature space of the classifier under the assumption that the learned representations follow conditional Gaussian distributions. **LogitNorm** (Wei et al., 2022) mitigates the over-confidence issue by training and testing with normalized logit vectors. **CIDER** (Ming et al., 2023) regularizes the model's hyperspherical space by increasing inter-class separability and intra-class compactness. **NPOS** (Tao et al., 2023) is a non-parametric version of VOS which removes the Gaussian assumption and instead adopts KNN to model the feature distribution.

**Training methods with outlier data.** While most methods consider the standard ID-only training, some works assume the access to auxiliary OOD training samples. **OE** (Hendrycks et al., 2019a) is the seminal work in this thread, which lets the classifier learn OOD detection in a supervised fashion. **MCD** (Yu and Aizawa, 2019) considers an ensemble of multiple classification heads and promotes the disagreement between each head's prediction on OOD samples. **UDG** (Yang et al., 2021a) proposes a clustering-based method to practically extract OOD samples from a mixed pool of auxiliary data and to improve the learned representation quality with unsupervised learning. **MixOE** (Zhang et al., 2023a) performs pixel-level mixing operations between ID and OOD samples and regularizes the model such that the prediction confidence smoothly decays as the input transitions from ID to OOD.

**Data augmentations**. We consider several data augmentation methods which have demonstrated success for improving the generalization ability of image classifiers. **StyleAugment** (Geirhos et al., 2019) applies style transfer to clean images to emphasize the shape bias over the texture bias. **RandAugment** (Cubuk et al., 2020) randomly sample the augmentation operation and magnitude to increase the diversity of augmented images. **AugMix** (Hendrycks et al., 2020) linearly interpolate between the clean and the augmented image to preserve the natural looking/fidelity of training images for better generalization. **DeepAugment** (Hendrycks et al., 2021a) manipulates the low-level statistics of clean images by sending them through image-to-image network and distorting the network's weights. **PixMix** (Hendrycks et al., 2021c) mixes two images with conical combination to create various new inputs with similar semantics. **RegMixup** (Pinto et al., 2022) trains the model with both clean images and mixed images obtained from convex combination.

## Appendix D. Generalizing OpenOOD to Other Tasks

In Section 2, we have discussed that the formal definition of OOD detection is general and can suit various tasks not limiting to classification. Here we investigate whether in practice the OpenOOD framework is applicable to other tasks.

We take an image regression task, more specifically the task of predicting age from face images, as an example. We use a pre-trained CNN-based age predictor model[11] and apply three simple post-hoc OOD detectors (more complex ones are also applicable but would require hyperparameter tuning). The OOD images are sampled from Textures (Cimpoi et al., 2014) and Places (Zhou et al., 2017). They are images of patterns and scenes, which are

---

11. `https://github.com/yu4u/age-estimation-pytorch`

obviously OOD w.r.t. human face distribution according to Definition 1. The pre-trained age estimator predicts a continuous age value by computing a weighted sum over discrete age values (0, 1, ..., 100), where the weights are determined by applying a softmax function over the raw model output. As a result, the structure of the model output is very similar to that in a classification task, making all methods supported in OpenOOD applicable to this regression problem.

|          | MSP   | MLS   | EBO   |
|----------|-------|-------|-------|
| Textures | 58.35 | 59.61 | 45.30 |
| Places   | 39.73 | 41.47 | 63.65 |

Table 5: OOD detection performance (AUROC) of three simple detectors on the age estimation problem. Notice that the random-guessing baseline is 50%.

We show the results in Table 5, where for each OOD dataset there always exist certain methods that achieve non-trivial detection rate (10% over the random-guessing baseline). Although the performance may not be extremely satisfying, the fact that we can readily apply multiple methods without modifications to perform OOD detection for this brand new task serves as strong evidence of the applicability of the OpenOOD framework. In fact, as long as the model output structure is similar to the classification model considered by current OpenOOD (which is indeed the case for various tasks including object detection, semantic segmentation, document classification, etc.), then OpenOOD is ready for use with minimal adaptation and changes required.

## Appendix E. Related Work

To the best of our knowledge, OpenOOD (especially the v1.5 release) is the only work that comprehensively benchmarks various OOD detection methods on multiple ID-OOD pairs. That said, there are still a few works that relate to OpenOOD in certain aspects.

Tajwar et al. (2021) made the observation that "OOD detection methods are inconsistent across data sets" from experiments on 3 small data sets (CIFAR-10, CIFAR-100, and SVHN) with 3 specific post-hoc methods (MSP, ODIN, and MDS). While we draw a similar conclusion of "no single winner" in Section 6, our observation comes from the experimentation with 4 data sets and nearly 40 methods from different categories.

A recent work by Galil et al. (2023) proposed a method for constructing OOD detection benchmark and evaluated the performance of 5 post-hoc methods with ImageNet-1K pre-trained models. Specifically, for a specific ImageNet-1K model with a specific post-processor, they consider ImageNet-21K images as OOD and categorizes OOD images into a sequence of difficulty groups based on the OOD score from the post-processor. Correspondingly, their evaluation looks at the OOD detection AUROC across all groups, intending to provide a spectrum of AUROC v.s. difficulty. We see two shortcomings of such practice for constructing a general benchmark. First, their benchmarking process is extremely time-consuming since it needs to iterate through nearly all of the samples in ImageNet-21K, which could be prohibitive even for the most lightweight method considering the compute required by common ImageNet models. Second, the resulting benchmark is diagnostic to both the classifier and the post-

processor. For example, the first difficulty group of the benchmark for MSP and that for ASH would *not* contain the same OOD samples, making the comparison ambiguous and much less straightforward. In comparison, our carefully designed benchmarks are *standardized*, *i.e.*, agnostic to classifiers and post-processors. Plus, we consider a wide range of methods beyond a few specific post-hoc approaches.

One work that most closely relates to ours is PyTorch-OOD (Kirchheim et al., 2022), which is a python library for evaluating OOD detection performance. There are several distinctions that separate OpenOOD from PyTorch-OOD. 1) **Number of supported methods.** PyTorch-OOD implements 19 methods as of May 2023 with the most recent one dating back to 2022, while OpenOOD supports 40 approaches including the most advanced ones published in 2023. 2) **Reliability of evaluation results.** PyTorch-OOD still includes as OOD images the LSUN-R and TIN-R (Liang et al., 2018) which contain obvious resizing artifacts (Tack et al., 2020). It also considers ImageNet-O which is known to cause biased evaluation since it is constructed by adversarially targeting a ResNet-50 model with the MSP detector (Galil et al., 2023). Their benchmarking results thus can be problematic and unreliable. 3) **Alignment between the goal reflected by the evaluation and human perception.** PyTorch-OOD's evaluation setup favors detectors that flag covariate-shifted ID samples (*e.g.*, those from ImageNet-R or ImageNet-C) as OOD. We argue that this does not align with human perception and is not an ideal behavior as thoroughly discussed in Section 2 and Section 6.

Another concurrent work by Bitterwolf et al. (2023) put up a new OOD data set (NINCO) for ImageNet-1K in response to the observed noise that exists in some earlier OOD data sets. They then evaluate 8 post-hoc methods on NINCO and specifically study the effect of large-scale pre-training. OpenOOD is inherently complementary to that work, as we intend to build a comprehensive benchmark for OOD detection by implementing and evaluating various types of methods (not restricting to post-hoc ones) on multiple data sets including ImageNet-1K. Meanwhile, our investigation on full-spectrum detection and findings regarding data augmentation techniques are unique.

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
