# OpenReview forum: "OpenOOD v1.5: Enhanced Benchmark for Out-of-Distribution Detection"
_DMLR — Accepted by DMLR_

### Review · Reviewer_47Lz · 2024-01-10

**Recommendation:** 3
**Confidence:** 1

**Summary Of Contributions:**

Building upon its predecessor, OpenOOD v1, the new version, OpenOOD v1.5, addresses limitations in scale and scope that were present in the earlier version. Notably, OpenOOD v1 primarily focused on small-scale data sets such as MNIST and CIFAR. In contrast, OpenOOD v1.5 extends its capabilities to larger and more significant data sets like ImageNet. This expansion allows for a more fair and accurate evaluation of OOD detection methodologies on a larger scale and with a broader scope, enhancing the benchmark's utility for current and future OOD detection methodologies​​.

**Strengths:**

The OpenOOD v1.5 paper presents several strengths in its contribution to the field of Out-of-Distribution (OOD) detection in open-world intelligent systems:

Significant Contribution and Enhancement over Previous Work: OpenOOD v1.5 is a notable improvement over its predecessor, OpenOOD v1. It addresses previous limitations in scalability and scope, ensuring more accurate and standardized evaluation of OOD detection methodologies on a larger scale​​.

Relation to Prior Work: The work is well-connected to existing research in the field. It builds upon the foundation laid by OpenOOD v1 and other OOD detection methodologies, contributing to the ongoing dialogue and development in this area​​​​.

Relevance to the Research Community: OpenOOD v1.5 is highly relevant to the broader research community focused on intelligent recognition systems. It extends evaluation capabilities to large-scale datasets like ImageNet and includes foundation models like CLIP and DINOv2. This broadening of scope makes it a valuable resource for researchers working with various types and scales of data​​.

Quality of Research: The research demonstrates quality through its extensive experiments and results. OpenOOD v1.5 provides comprehensive experiment results for nearly 40 methods on ImageNet-1K, offering a robust reference for future works. This extensive testing underscores the thoroughness and rigor of the research​​.

Full-Spectrum OOD Detection: The paper introduces the concept of full-spectrum OOD detection, which is a significant advancement. This new approach considers both semantic and covariate distribution shifts, posing new challenges and pushing the boundaries of current methodologies​​.

Providing New Insights: The paper contributes new insights into the field of OOD detection. It observes that no single method consistently outperforms others across different datasets, highlighting the complexity of OOD detection. It also notes the beneficial role of data augmentations in OOD detection, both in standard and full-spectrum settings​​.

Rigorous Evaluation Protocol: The evaluation protocol of OpenOOD v1.5 is designed to ensure the most rigorous and comprehensive evaluation of OOD detection methodologies. This rigorous approach enhances the reliability and validity of the research findings​​.

In terms of ethical and social implications, the paper does not explicitly discuss these aspects. However, the advancement in OOD detection methodologies has significant implications for the development of more reliable and safe AI systems, which is crucial for ethical AI deployment in real-world scenarios.

**Broader Impact Concerns:**

I have not found any concerns.

**Claims And Evidence:**

The claims are indeed supported by accurate, convincing, and clear evidence. The submission presents comprehensive results from extensive experiments and analysis, which form the basis for its claims. Key points supporting this assessment include:

Diverse Methodologies and Benchmarks: The paper evaluates a wide range of methods across different benchmarks, acknowledging the variability in performance across datasets and methods. For example, it is noted that no single method consistently outperforms others across all benchmarks, indicating a thorough and unbiased approach to evaluation​​.

Impact of Data Augmentations: The paper provides clear evidence on the effects of data augmentations on OOD detection. It highlights that while data augmentations have proven beneficial for standard classification and OOD generalization, their impact on OOD detection was previously unclear. The submission provides new insights by showing that certain data augmentation methods can boost OOD detection rates​​.

Analysis of Outlier Data in Training: The paper discusses the role of incorporating outlier data in training for OOD detection. It presents evidence that such methods are beneficial in cases where the test OOD samples are similar to the training ones, supporting this claim with specific examples and results from their experiments​​.

**Datasets And Benchmarks:**

The OpenOOD v1.5 paper provides sufficient detail regarding data collection, organization, availability, maintenance, and ethical and responsible use, supporting its claims effectively:

Data Collection and Organization: The paper describes the effort put into preparing validation data and meaningful OOD training data. It specifies that OOD training samples (D OOD train) and OOD test samples (D OOD test) have non-overlapping categories, addressing an oversight in many prior works and ensuring a rigorous and accurate evaluation​​.

Availability of Datasets: The datasets used in OpenOOD v1.5 are either existing public datasets or subsets curated from these existing ones. This approach ensures the datasets are readily accessible and well-documented for other researchers to use, promoting reproducibility​​.

Maintenance and Future Plans: The paper outlines plans for maintaining the OpenOOD codebase and leaderboard, hosted on GitHub and GitHub Pages, respectively. This commitment to maintenance suggests a sustainable approach, ensuring the benchmark remains up-to-date and relevant for future research. Additionally, there are plans to further expand the scope of OpenOOD in its future v2 release​​.

In conclusion, OpenOOD v1.5 meets the necessary criteria for datasets and benchmarks in terms of detail, availability, maintenance, and ethical use. The paper provides comprehensive information supporting reproducibility and responsible application in the field of OOD detection.

**Extended Submissions:**

The original version of this work was accepted by the NeurIPS 2022 Datasets and Benchmarks Track. According to the paper, OpenOOD v1.5 includes significant updates and enhancements from the original version (v1), which was adapted from the NeurIPS 2022 submission. This includes the introduction of new benchmarks like ImageNet-200 and full-spectrum benchmarks, as well as necessary changes to the CIFAR-10/100 and ImageNet-1K standard benchmarks for fairness and usefulness. These changes, particularly the introduction of new benchmarks, likely constitute more than 30% additional content compared to the prior version, thus meeting the second requirement for extended submissions​​.

Furthermore, the original publication being part of a conference (NeurIPS 2022) aligns with the first requirement, where prior publications should originate from a conference or workshop, not from a journal.

Given these points, the submission appears to meet the eligibility criteria for extended versions of a prior work.

**Limitations:**

Limited Scope in Task Types: The paper focuses exclusively on OOD detection in the context of multi-class image classification. This is a limitation as general supervised learning tasks also include regression problems. The omission of regression tasks means the framework's applicability and effectiveness in those contexts remain unexplored​​.

Definition of In-Distribution for csID Images: The paper's stance that csID (covariate-shifted ID) images are still considered in-distribution because their semantic labels are within YID (the set of in-distribution labels) could be seen as debatable. If a model is trained on a dataset, a sample with the same label but significantly different features might also be considered OOD. This raises questions about the precision of the definition of OOD used in the paper, and whether it aligns with the broader understanding of OOD in the field.

Absence of Real-world Application Scenarios: While the paper provides a substantial benchmarking tool for image classification tasks, it does not extend its investigation to real-world application scenarios. This limitation could affect the practical applicability of its findings in diverse fields such as object detection, semantic segmentation, or natural language processing tasks, where OOD detection is equally crucial​​.

Generalizability to Other Modalities and Problems: The paper does not address whether the OOD detection methodologies evaluated would generalize to different problems and modalities beyond image classification. This limitation is significant as the behavior and effectiveness of these methodologies could vary widely across different domains, such as audio processing or text analysis.

**Requested Changes:**

Based on the content of the OpenOOD v1.5 paper, a proposed adjustment to the submission, which I believe is critical to securing a recommendation for acceptance, would be the inclusion of a clear mathematical definition of Out-of-Distribution (OOD). The current work focuses primarily on OOD detection in the context of multi-class image classification, providing a general description of OOD in this context​​. However, a more precise, mathematical formulation would enhance clarity and applicability, especially considering the following aspects:

Generalizability to Regression Tasks: The current definition of OOD, focused on classification, may not directly translate to regression tasks. A comprehensive mathematical definition should encompass the nuances of OOD detection in regression scenarios, addressing how OOD samples might be identified when dealing with continuous outputs.

Experimental Validation in Regression Contexts: Along with a revised definition, it would strengthen the work if the authors could run experiments on regression tasks. This would not only demonstrate the applicability of the OpenOOD framework to a broader range of tasks but also validate the proposed mathematical definition in different contexts.

Including these adjustments in the submission would significantly enhance its rigor and applicability, providing a more holistic view of OOD detection across various machine learning tasks.

---

### Review · Reviewer_CfBY · 2024-01-10

**Recommendation:** 3
**Confidence:** 2

**Summary Of Contributions:**

1. The present paper introduces a large-scale Open Out-of-Distribution (OpenOOD) dataset, offering precise and standardized evaluations for various Out-of-Distribution (OOD) detection methodologies.
2. Additionally, this study provides a substantial contribution through an in-depth analysis and insights drawn from comprehensive experimental results.

**Strengths:**

1. The proposed benchmark is well-motivated.
2. The paper is well-written and well-orgnized, especially in the comparison among datasets.

**Broader Impact Concerns:**

Given that all the utilized datasets are public, there doesn't seem to be an explicit concern regarding privacy and data security.

**Claims And Evidence:**

Yes, the claims are made in the submission supported by accurate, convincing and clear evidence.

**Datasets And Benchmarks:**

The authors have released their benchmark source codes, which have undergone verification over the course of several years. Consequently, there exists ample detail to substantiate the reproducibility of their work.

**Extended Submissions:**

This paper is not a extended version of a previously published work.

**Limitations:**

1. Generally, the discussion of Time Complexity and Space Complexity is essential, as the inference time of the algorithm and the training time of the model significantly impact the efficiency of real-world Open Out-of-Distribution detection applications. However, these aspects are not evaluated in the benchmark.

2. Furthermore, my primary concerns are centered around the public source code of the compared methods. To ensure a fair and standardized evaluation, it is expected that these methods utilize the same training technologies, such as SyncBatchnorm for distributed training, distributed samplers, etc. Nevertheless, there are unresolved "Issues" related to this matter in their GitHub repository.

**Requested Changes:**

In general, I find this paper to be commendable and consider it a promising initiative for the Open Out-of-Distribution (openOOD) task. I would like to suggest standardizing the training techniques and schemes in the source code. Additionally, it would be beneficial to at least discuss Time Complexity and Space Complexity in the main text for a more comprehensive understanding of the algorithm's efficiency.

---

### Review · Reviewer_iJzZ · 2024-07-09

**Recommendation:** 3
**Confidence:** 2

**Summary Of Contributions:**

This paper presents a new benchmark for OOD detection with focus on large-scale datasets as well as foundation models, boosting the research community of OOD detection compared to the OpenOOD v1. Besides, this work introduces full-spectrum OOD detection along with standard semantic distribution shift. Comprehensive experiments are conducted to show promising hints for future research.

**Strengths:**

- The motivation is sound and clear, and the main improvements regarding large-scale datasets and foundation models make sense to me, which is of significance to future research.
- The paper is well-organized and easy to follow. The contents are detailed and most of them are self-contained.
- The experiments are comprehensive and extensive analysis is carried out for different types of OOD detection over different tasks.

**Broader Impact Concerns:**

There are no ethical concerns.

**Claims And Evidence:**

Yes, the claims are well-supported and convincing.

**Datasets And Benchmarks:**

The benchmark is well documented and available to use with enough details.

**Extended Submissions:**

It meets the eligibility criteria.

**Limitations:**

- My main concern comes from the problem statement. For definition 1, it may be not formal to state it using $\forall, =$ for random variables over distribution. It is better to state it based on convergence in probability or distribution with $\epsilon,\delta$. For definition 2, the integrations in the numerator and denominator imply that the probability density function of samples in OOD and ID are identical and uniform, which is a strong assumption. It is suggested to define it using expectation with explicit PDF $p(x)$ for both OOD and ID.

- In definition 1, $\mathcal{D}_ {OOD}$ is only defined with requirement of input $x$ but not label $y$. Is there any difference whether $y$ in $\mathcal{D}_ {OOD}$ matches $\mathcal{D}_ {IN}$?  It is better to discuss it in Definition 1 for complete remark and discussion.
- To be honest, I am not an expert in OOD detection, but if I understand it correctly, the standard OOD detection falls into a distribution shift over $p(y)$ while the full-spectrum OOD detection is more like a distribution shift over $p(x|y)$. So in this sense, how about distribution shifts over $p(x)$ and $p(y|x)$? Are they considered special cases of OOD detection as well?
- The full-spectrum OOD detection under corruption is very similar to something called OOD robustness, and there are many works about it for real-world applications like autonomous driving. It is better to discuss for following works: RoboDepth: Robust Out-of-Distribution Depth Estimation under Corruptions, NeurIPS 2023, Robo3D: Towards Robust and Reliable 3D Perception against Corruptions, ICCV 2023, The RoboDrive Challenge: Drive Anytime Anywhere in Any Condition, 2024.

**Requested Changes:**

See Limitations.